# Nasal DNA methylation at three CpG sites predicts childhood allergic disease

Merlijn van Breugel[1,2,3,18], Cancan Qi[1,2,18], Zhongli Xu[4], Casper-Emil T. Pedersen[5], Ilya Petoukhov[3], Judith M. Vonk[2,6], Ulrike Gehring[7], Marijn Berg[2,8], Marnix Bügel[3], Orestes A. Carpaij[2,8], Erick Forno[4], Andréanne Morin[9], Anders U. Eliasen[5,10], Yale Jiang[4], Maarten van den Berge[2,11], Martijn C. Nawijn[2,8], Yang Li[12,13,14], Wei Chen[4], Louis J. Bont[15,16], Klaus Bønnelykke[5], Juan C. Celedón[4], Gerard H. Koppelman[1,2,19] & Cheng-Jian Xu[12,13,14,17,19] ✉

Childhood allergic diseases, including asthma, rhinitis and eczema, are prevalent conditions that share strong genetic and environmental components. Diagnosis relies on clinical history and measurements of allergen-specific IgE. We hypothesize that a multi-omics model could accurately diagnose childhood allergic disease. We show that nasal DNA methylation has the strongest predictive power to diagnose childhood allergy, surpassing blood DNA methylation, genetic risk scores, and environmental factors. DNA methylation at only three nasal CpG sites classifies allergic disease in Dutch children aged 16 years well, with an area under the curve (AUC) of 0.86. This is replicated in Puerto Rican children aged 9–20 years (AUC 0.82). DNA methylation at these CpGs additionally detects allergic multimorbidity and symptomatic IgE sensitization. Using nasal single-cell RNA-sequencing data, these three CpGs associate with influx of T cells and macrophages that contribute to allergic inflammation. Our study suggests the potential of methylation-based allergy diagnosis.

Allergic diseases such as asthma, rhinitis, and eczema are highly prevalent, non-communicable childhood diseases worldwide[1] that have a significant impact on quality of life and cause a considerable burden on healthcare systems[2,3]. Allergic diseases often coexist in the same individual, suggesting shared underlying mechanisms[4]. The prevalence of allergic diseases has increased rapidly for more than 50 years, and 40–50% of schoolchildren in the Western world have a positive allergen-specific Immunoglobulin (IgE) to one or more common allergens[5]. However, a positive allergen-specific IgE does not necessarily coincide with presenting allergy symptoms, and effective biomarkers are needed to capture the allergic inflammation signal rather than the presence of IgE. There is, therefore, a great need for non-invasive biomarkers to facilitate better diagnosis, especially in early childhood.

Childhood allergic diseases have strong shared genetic and environmental components[6,7]. Many shared genetic risk variants for

allergic disease influence the expression of immune-related genes[8]. The increase in the prevalence of allergic diseases indicates that environmental exposures, such as microbial stimulation[9], air pollution, breastfeeding, pets at home, and cigarette smoking, also have an important influence[10]. Environmental factors can have sustained effects on gene expression through the modulation of epigenetic features such as DNA methylation[11]. Indeed, epigenome-wide association studies (EWAS) of allergic diseases in both whole blood and nasal-brushed cells have shown many shared epigenetic signatures of allergy[12–15].

There is a need for the prediction of individual allergic disease risk, especially in young, preschool children, in whom allergy is difficult to diagnose. While previous studies have constructed models to assess individual disease risk, these models investigated usually only a single layer of 'omics' data, such as the genome, epigenome, transcriptome, proteome, or metabolome[14,16–19]. Given the complex

biological nature of allergic diseases, with the known involvement of multiple omics layers and environmental factors, leveraging the information from multi-omics data may improve the prediction of these diseases[20]. Recently, an integrated multi-omics approach was successfully used to predict cytokine production of immune cells[21], but there are few complex disease prediction models using multi-omics approaches.

In this study, we develop a machine learning prediction model for allergic disease, by systematically integrating large-scale multi-omics data from the genome, DNA-methylome of blood and nasal cells, and environmental factors. We first evaluate the predictive power of each layer separately and identify the most predictive features to construct a parsimonious allergy prediction model in the Dutch PIAMA (Prevention and Incidence of Asthma and Mite Allergy) birth cohort[22]. We replicate our model in three independent birth cohorts of children of different ethnicities and ages. We report that allergy can be predicted accurately using nasal DNA methylation at only three CpG sites. These sites also detect allergic multimorbidity and symptomatic sensitization, and can be related to the influx of T cells and macrophages in the nasal mucosa.

## Results

### Nasal DNA methylation has the most predictive power for allergic disease

To construct a multi-omics prediction model of allergic disease, we first investigated the relative contributions of each data layer, including genomics, blood and nasal DNA methylation, and personal and environmental factors. We performed these analyses in the PIAMA cohort, a prospective national population-based birth cohort study on the development of asthma and allergy in the Netherlands[22] (Fig. 1a). Here, the allergic disease was defined by both the presence of asthma/rhinitis/eczema and the detection of allergen-specific IgE in blood; we found a prevalence of 19.3%.

In 348 children aged 16 years with complete data (Table 1), we selected features from four data layers (genome, blood, nasal methylome, and environmental factors) that had been previously associated with allergic disease (see Methods). In total, we analyzed 467 features. We evaluated six machine learning methods (see Methods) and adopted Elastic Net[23] as our model of choice because it offered accurate performance, low overfit, and good interpretability (Supplementary Table 1).

The largest predictive power was attributed to DNA methylation levels in nasal epithelial cells (Fig. 2a); other layers had a negligible effect. This conclusion did not change when we varied the order in which the layers were added. 86% of the top 50 features were nasal and 14% were blood DNA methylation CpG sites (Supplementary Table 2), underlining the predictive power of the DNA-methylome. These results indicate that genetic factors had very limited predictive power for the allergic disease at the individual level, which was also observed in a larger dataset of 675 individuals from the PIAMA cohort for whom both genotype and allergy phenotype data were available (Supplementary Fig. 1).

### A parsimonious prediction model with only three nasal CpG sites predicts allergy

We next aimed to generate a parsimonious model to increase our model's robustness, reproducibility, and cost-effectiveness, and to facilitate translation to a clinical setting (Fig. 1b), starting with the top-ten ranked features (Fig. 2b), all of which were nasal DNA CpG methylation sites (Supplementary Table 3). When we compared the performance of models with an incremental number of features, we observed that after including the first three nasal CpG sites, additional CpG sites did not further improve model performance ($p$ value = 0.76, two-sided corrected repeated k-fold cross-validation (cv) test[24], Fig. 2c). The other test statistics are displayed in

Supplementary Table 4. These three CpGs (cg20372759, cg01870976, and cg24224501) are located near gene *CYP27B1*, *SNRPA1*, and *LRRC17* respectively (Supplementary Table 5). The "three-CpG sites" model performed well in predicting allergic disease in the PIAMA discovery cohort, with an ROC AUC (receiver operating characteristic, area under the curve) test score of 0.86 (Fig. 3a). This parsimonious model has a near-zero overfit (ROC AUC difference of 0.005), as the ROC AUC training score was also 0.86 (Fig. 3a). Because of the imbalance of cases and controls in the discovery dataset, we also introduced the precision-recall curve (PRC)[25] to evaluate model performance: the PRC AUC of the model was 0.50 (Supplementary Fig. 2a). The model's coefficients can be found in Supplementary Table 6. The three-CpG model was also compared to the previously published nasal 30-CpG model that performed well in predicting atopy (i.e., the presence of IgE sensitization) in the Epigenetic Variation and Childhood Asthma in Puerto Ricans study (EVA-PR)[14]. Although we only used three nasal CpG sites, our model showed a comparable performance ($p$ value = 0.32, two-sided corrected repeated k-fold cv test) with the more complex 30-CpG model (ROC AUC test score of 0.83) (Supplementary Fig. 3a). Overfit of this more complex model remained low (ROC AUC difference of 0.008), albeit slightly larger than our three-CpG model. The PRC AUC of 0.48 was also lower in the 30-CpG model (Supplementary Fig. 3b), although the difference was not significant ($p$ value = 0.40, two-sided corrected repeated k-fold cv test).

### Three-CpG model also predicts allergic disease in Puerto Rican children

To test its generalizability across ethnicities, we first replicated our three-CpG model in a cohort of comparably aged Puerto Rican children from Puerto Rico (EVA-PR)[14] (Fig. 1c). In EVA-PR, the prevalence of allergic disease was 55.5% and the mean age was 15.5 years (Table 1). Our model performed well on both ROC AUC (0.82) (Fig. 3b) and PRC AUC (0.84) (Supplementary Fig. 2b). The PRC AUC's baseline value (of a random model) is equal to the disease prevalence, here 0.56, which explains its step increase relative to the discovery cohort. The good replicability is affirmed by the precision and recall metrics of 0.80 and 0.79, respectively, for the EVA-PR cohort, compared to 0.53 and 0.64 in the PIAMA cohort (Supplementary Table 7). We also performed the replication on stratified age groups in EVA-PR (Supplementary Fig. 4), which showed that model performance is very good at similar (15–17 years) and higher ages (18–20 years); and remains satisfactory for the age group 9–14 year (Supplementary Fig. 4).

We next extrapolated our model to two cohorts of younger children (6-year-olds; COPSAC2010, Denmark and MAKI trial[26], the Netherlands, Table 1), both with an ROC AUC of 0.79 (Fig. 3c, d). Our model's performance in the younger cohorts showed considerably lower recall (Supplementary Fig. 2c, d), especially in MAKI, where recall dropped to 0.03 (Supplementary Table 7). This indicates we should be cautious in extrapolating our prediction model to this younger age group. In order to understand the difference in performance between age groups, we compared the probe distribution of the three CpG sites in participants with and without allergic disease in all four discovery and replication cohorts (Fig. 3e–h). We observed that the DNA methylation levels of the three CpG sites were clearly lower in participants with allergic disease than in participants without allergic disease in all cohorts, with the highest difference seen in adolescents and paralleled by the better prediction power in these two cohorts. The DNA methylation levels in control samples were higher in the younger cohorts compared to the adolescent cohorts, which may be attributed to the ageing effect on DNA methylation (Fig. 3e–h).

### DNA methylation at three CpG sites relates to symptomatic IgE sensitization and disease multimorbidity

The allergic disease was defined as a combination of diseases (asthma/rhinitis/eczema) and IgE sensitization. To better understand the

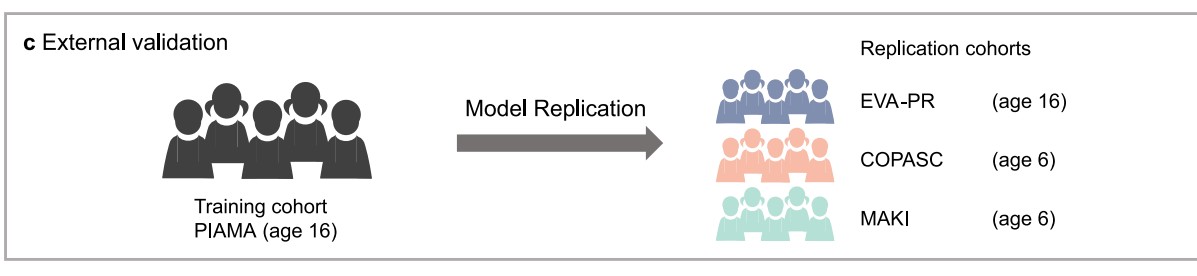

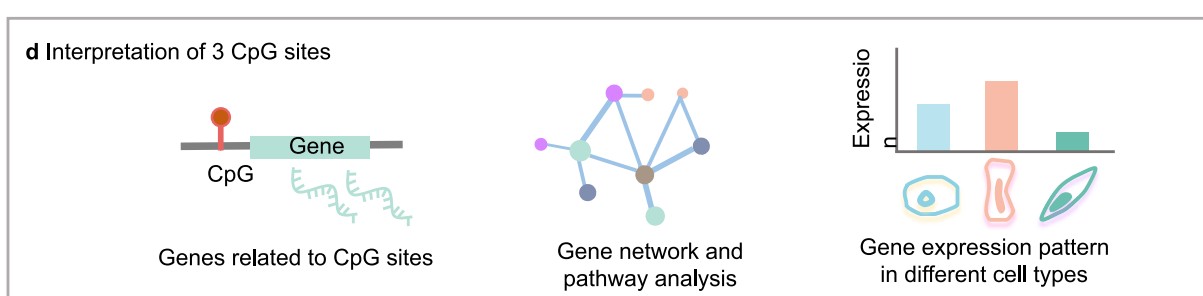

association of the three top CpG sites with both disease and IgE sensitization, we performed group-by-group comparisons in the discovery cohort, analyzing symptomatic and asymptomatic children with and without IgE sensitization. The results showed that these CpG sites were not only associated with IgE sensitization, but also with the presence of clinical symptoms in IgE-sensitized participants (Fig. 4a). Importantly, we observed significant differences between the IgE+ symptom+ group and the IgE+ symptom− group for the three CpG sites ($p$ values = $1.4 \times 10^{-3}$, $6.2 \times 10^{-4}$, and $9.8 \times 10^{-4}$, for CpG cg01870976, cg20372759, and cg24224501, respectively, two-sided Student's $t$-test), suggesting that these three sites distinguish symptomatic from asymptomatic sensitization. Next, we found lower DNA methylation of these CpG sites in multimorbidity compared to a single allergic disease ($p$ values = 0.039, 0.047, and 0.17, two-sided Student's

**Fig. 1 | Study design to develop and validate the prediction model and functionally interpret its predictors. a** We integrated multi-omics data, including, genetics, DNA methylation from blood and nasal brushes, and environmental factors from the PIAMA cohort. Using machine learning methods, we aimed to (1) assess the contribution of each data layer to the performance of a prediction model; and (2) present a simple prediction model for allergic disease. **b** To develop a parsimonious prediction model, we first ranked all features from the full dataset, then features were added incrementally until no significant increase in performance was seen. The performance of the final parsimonious model was demonstrated in the discovery cohort. **c** The final model was evaluated by applying it to another similar but independent cohort (EVA-PR) and two independent younger children cohorts, COPSAC2010 and MAKI. **d** To understand the information that was captured by the three CpG sites used in the model, we linked the methylation level of the CpG sites to the expression of genes by eQTM analysis. The eQTM genes were functionally interpreted by gene network and pathway analysis; scRNA-seq data were used to interpret the expression pattern of eQTM genes in different nasal cell types.

## Table 1 | Characteristics of the discovery and replication cohorts

| | PIAMA (Discovery) | EVA-PR (Replication) | COPSAC2010 (Replication) | MAKI (Replication) |
|---|---|---|---|---|
| Sample size | 348 | 481 | 503 | 259 |
| Age (years) | 16.3 (0.20) | 15.5 (3.0) | 6.0 (0.2) | 5.8 (0.4) |
| Gender male | 179/348 (51.4%) | 248/481 (51.6%) | 217/503 (43.1%) | 146/259 (56.4%) |
| Smoking * | 37/297 (12.5%) | NA | 35/ 503 (6.9%) | 15/259 (5.8%) |
| Asthma | 23/348 (6.6%) | 236/481 (48.9%) | 31/503 (6.2%) | 20/255 (7.8%) |
| Rhinitis | 59/348 (17.0%) | 295/481 (61.3%) | 60/503 (53.3%) | 27/259(10.4%) |
| Eczema | 29/348 (8.3%) | 89/481 (18.5%) | 55/503 (10.8%) | 28/259 (10.8%) |
| Positive allergen-specific IgE | 162/348 (46.6%) | 311/481 (64.7%) | 85/503 (18.7%) | 56/ 205 (27.3%) |
| Allergy | 67/348 (19.3%) | 267/481 (55.5%) | 37/503 (8.9%) | 29/259 (11.2%) |

Continuous variables are presented as mean (SD); categorical variables are presented as (number of "yes")/(total number (with this phenotype available)) (percentage).
*PIAMA* prevention and incidence of asthma and mite allergy birth cohort (Netherlands), *EVA-PR* epigenetic variation and childhood asthma in Puerto Ricans (USA), *COPSAC2010* Copenhagen prospective study on asthma in childhood (Denmark), *MAKI* trial (the Netherlands).
*Smoking in COPSAC2010 and MAKI cohort refers to second-hand smoking.

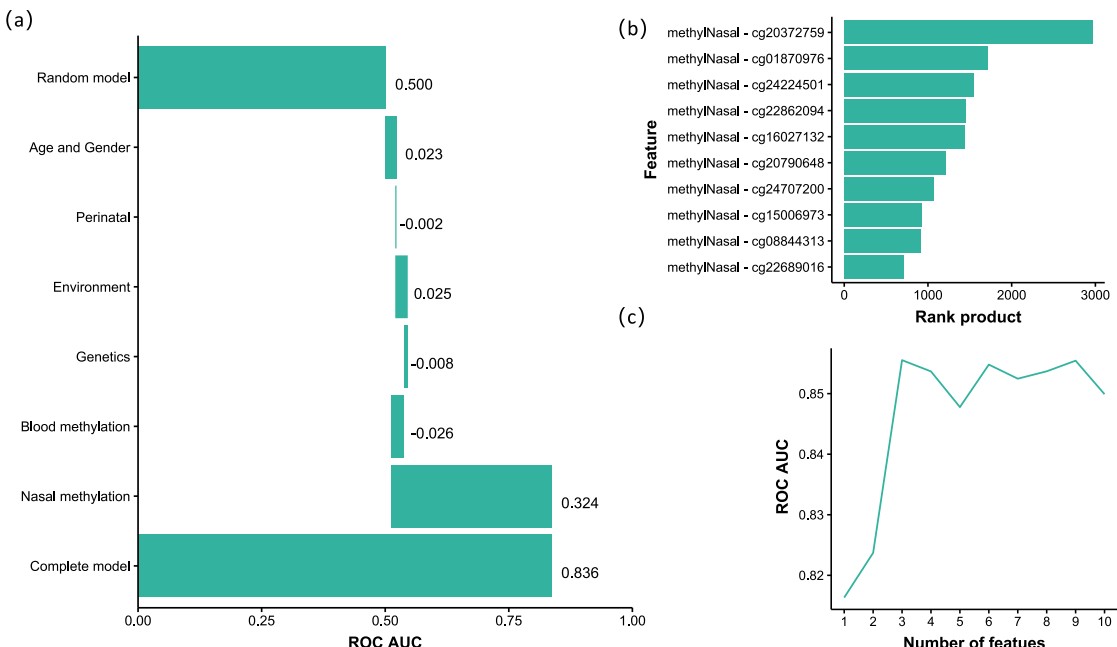

**Fig. 2 | Feature selection of the data layers and features that contribute most to model performance. a** Based on the validation performance of an Elastic Net model using a ten-times repeated tenfold cross-validation procedure. Layers were added sequentially: age and gender were added first, then perinatal factors were included in the model, etc. Negative contributions could occur when variables had low predictive power and increased model overfit. Perinatal features were low birth weight and breastfeeding; Environment: pets during pregnancy, maternal smoking, and older siblings at home; Genetics: allergy SNPs and polygenic risk scores (PRS) for the combined allergy phenotype, as well as for asthma, rhinitis, eczema, and IgE sensitization; Blood methylation: 219 CpG sites from blood cells previously associated allergy; Nasal methylation: 134 CpG sites from nasal cells previously associated with allergy. **b** Rank product of top-ten features. **c** The AUC of models with an incremental number of features; after the first three nasal CpG sites, adding another CpG site did not further increase the model's performance.

*t*-test, Fig. 4b). These findings were also exactly replicated in the EVA-PR cohort (Fig. 4c, d), confirming that nasal DNA methylation captures differences between asymptomatic and symptomatic sensitization and allergic disease multimorbidity.

## Genes associated with the three CpG sites identified a signature of immune cells

To interpret the biological changes responsible for the differential methylation levels of the three nasal CpG sites in allergy

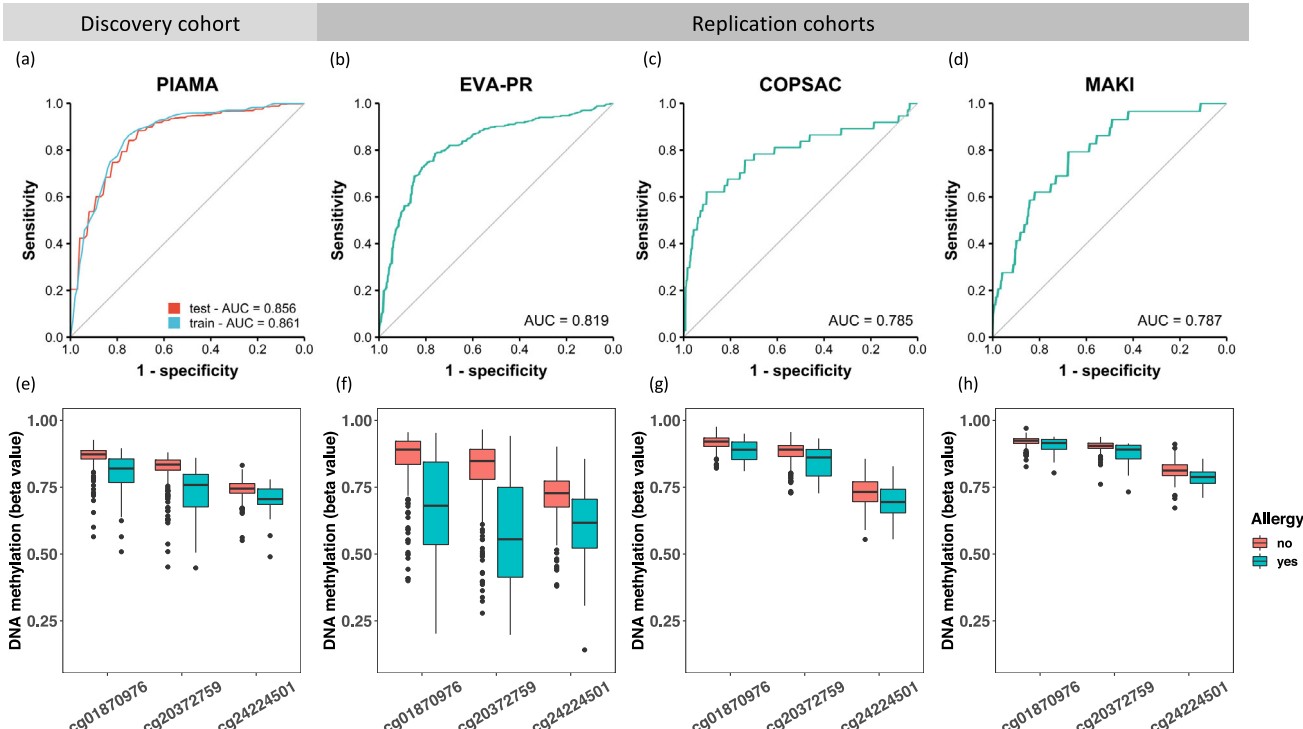

**Fig. 3 | Model performance and feature distribution in the discovery and replication cohorts.** The ROC curve and AUC of the model in the discovery cohort (**a**) and replication cohorts: EVA-PR (**b**, mean age 16 years), COPSAC2010 (**c**, mean age 6 years), and MAKI (**d**, mean age 6 years). The DNA methylation levels of the three CpG sites in participants with and without allergy are shown in (**e**–**h**), for PIAMA (**e**, $n = 348$), EVA-PR (**f**, $n = 481$), COPSAC2010 (**g**, $n = 503$), and MAKI (**h**, $n = 259$), respectively. Box plots show medians and the first and third quartiles (the 25th and 75th percentiles), respectively. The upper and lower whiskers extend the largest and smallest value no further than $1.5 \times$ IQR, with individual dots indicating individual observations outside this range.

(Supplementary Table 5), we first correlated DNA methylation to gene expression levels by expression quantitative trait methylation (eQTM) analysis, using matched nasal RNA-seq data from children of the PIAMA cohort ($n = 244$ with both types of data available). Expression levels of 127 unique genes (eQTM genes) were associated with methylation levels at the three CpG sites, resulting in 182 CpG-gene pairs (Supplementary Data 1), with many genes related to more than one of the three CpG sites (Supplementary Fig. 5). We applied weighted correlation network analysis (WGCNA[27]) on these eQTM genes and identified two modules (Supplementary Table 8). Using KEGG pathway enrichment analysis, we found that genes from Module 1 were enriched in ten, mostly T cell-related pathways, including hematopoietic cell lineage, Th1 and Th2 cell differentiation, and Th17 cell differentiation (Supplementary Fig. 6a). Genes from Module 2 were enriched in eight pathways (Supplementary Fig. 6b), including the B cell receptor signaling pathway and complement and coagulation cascades. These results suggest that these eQTM genes are related to immune cells and may be involved in the allergic inflammatory response.

We hypothesized that the differences in CpG methylation in nasal brushes in allergic and non-allergic participants reflected differences in cell type composition or different DNA methylation profiles in different cell types, with the eQTM genes reflecting the cell-type specific transcriptional profile. To map these putative cell types, we generated a single-cell RNA-seq dataset from nasal brushings of nine adult participants (five healthy and four asthmatics, Supplementary Table 9) to assess the cell-type specific expression pattern of the eQTM genes. We identified ten cell types: eight epithelial cell clusters (basal, goblet 1, goblet 2, squamous, cycling, ciliated, deuterosomal, and ionocytes), and two immune cell clusters (myeloid and T cells) (Fig. 5a). Most of the eQTM genes from Module 1 were highly expressed in the T cell cluster, while the genes from module 2 were enriched in the myeloid cell cluster (Fig. 5b). These findings were consistent when using another nasal Single-cell RNA-seq dataset[28] (Supplementary Fig. 7) which used the nasal polyp tissues that were different from nasal brushing cells used in this study. Subsetting and re-clustering of all immune cells (Fig. 5c) indicated the presence of five myeloid cell clusters (dendritic cells (DC) −1, DC-2, macrophages, classical monocytes and plasmacytoid dendritic cell (pDCs), Fig. 5e), 3 T cell clusters (CD8 T cells/cytotoxic T lymphocytes (CTLs), CD4 cells/Treg cells, and NK cells, Fig. 5f), and single cluster representing B cells, plasma cells, and mast cells. Genes from Module 1 were most highly expressed in CD8/ CTLs and CD4/Treg cells (Fig. 5d, h), while genes from Module 2 were enriched in macrophages (Fig. 5d, f), suggesting that the three CpG sites may capture the relative increase in both T cells and macrophages in allergic inflammation in nasal brushes.

### Allergy-associated SNP may mediate allergy through DNA methylation of a nasal CpG site

We performed methylation quantitative trait locus (MeQTL) analysis to investigate if the three nasal CpG sites are regulated by allergy-associated SNPs. For this, we associated the three CpG sites with SNPs previously associated with allergic disease[8]. We identified 15 SNP-CpG pairs showing nominal significance ($p$ value <0.05, Supplementary Table 10), suggesting potential regulation of genetic variants at the methylation level of these CpG sites. The most significant SNP was also associated with allergic disease in our PIAMA dataset ($p$ value = 0.022, Supplementary Table 11). We assessed the mediation effect of this SNP, located in the *ATG5* (*Autophagy Related 5*) gene, on the allergic disease through DNA methylation and found that 75.7% of the effect of rs9372120 on the allergic disease was mediated by DNA methylation of cg20372759 (Supplementary Table 11).

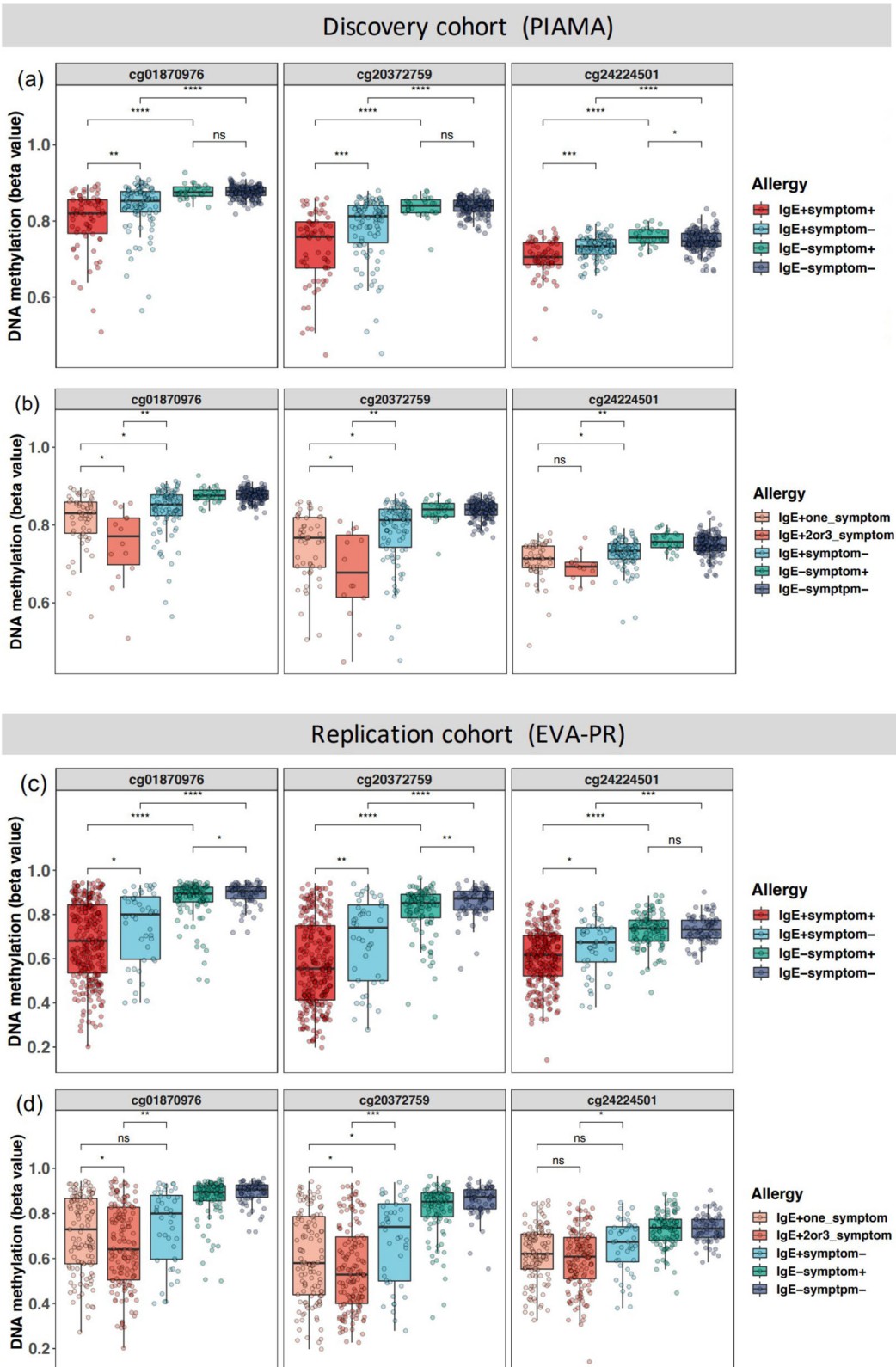

**Fig. 4 | DNA methylation levels of three CpG sites in subgroups of allergy phenotype.** Methylation levels are shown for the discovery (**a**, **b**, PIAMA, $n = 348$) and a replication cohort (**c**, **d**, EVA-PR, $n = 481$). Samples were stratified by both IgE sensitization and symptom (**a**, **c**), and were additionally stratified by the number of symptoms (**b**, **d**). The Student's t-test (two-sided) was used to compare the difference between different groups (*$P < 0.05$; **$P < 0.01$, ***$P < 0.001$, ****$P < 0.0001$, ns not significant). Box plots show medians and the first and third quartiles

(the 25th and 75th percentiles), respectively. The upper and lower whiskers extend the largest and smallest value no further than $1.5 \times$ IQR. Exact p values within PIAMA from top to bottom per CpG are as follows. a cg01870976: $4.6 \times 10^{-8}$, $1 \times 10^{-10}$, 0.89, 0.0014; cg20372759: $4.2 \times 10^{-9}$, $3 \times 10^{-12}$, 0.85, .00062; cg24224501: $2.4 \times 10^{-5}$, $9.4 \times 10^{-11}$, 0.029, .00098. **b** cg01870976: 0.0058, 0.031, 0.039; cg20372759: 0.0041, 0.014, 0.047; cg24224501: 0.0022, 0.012, 0.17.

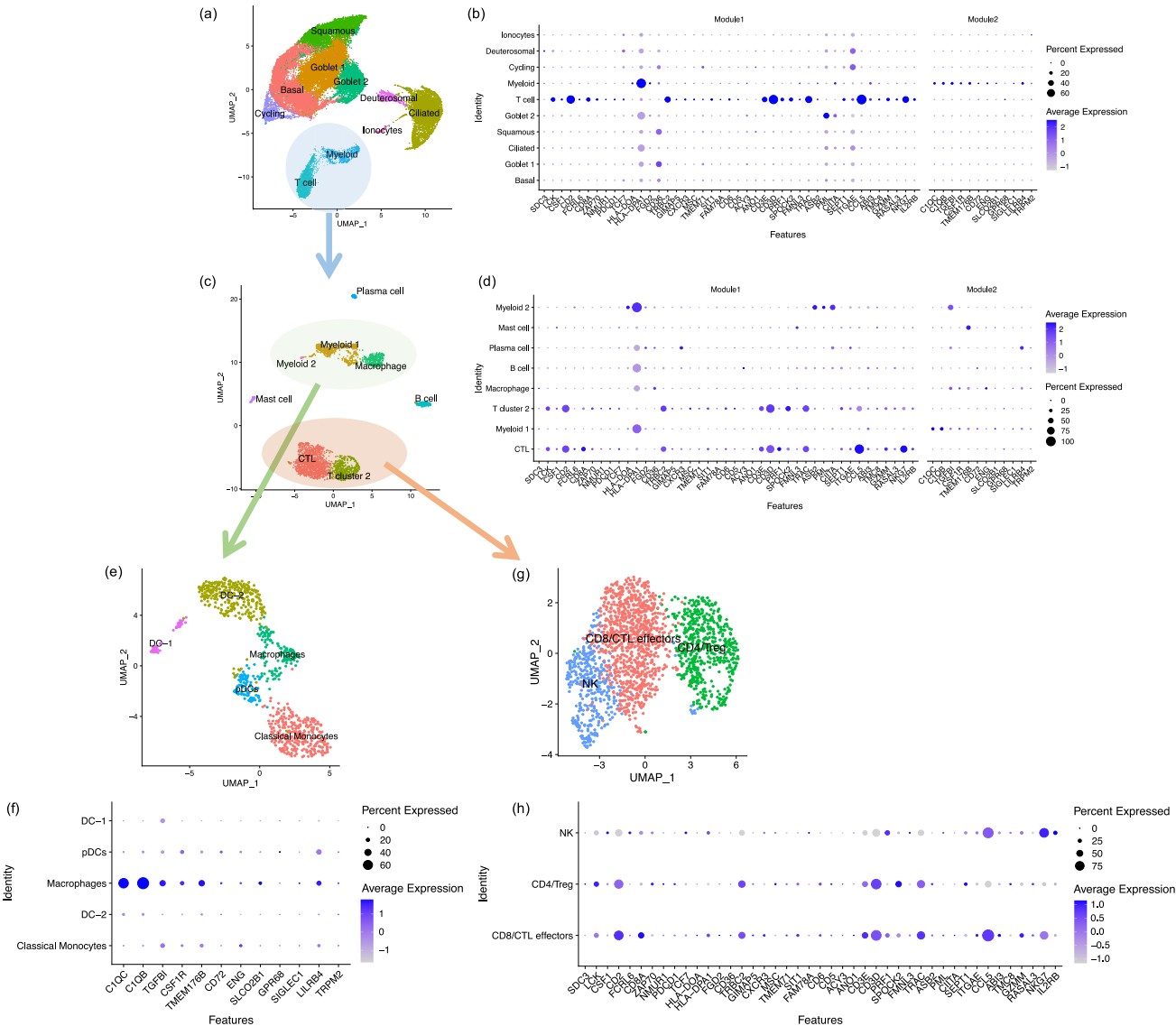

**Fig. 5 | Expression patterns of genes associated with three CpG sites in different cell types.** Using single-cell data, we identified ten cell types in nasal brushing cells (**a**). We identified two gene modules from eQTM genes by Weighted Gene Co-expression Network Analysis (WGCNA). Plot (**b**) depicts the average expression levels per cell cluster of genes from Module 1 and Module 2, which were available in the nasal scRNA-seq dataset in all cell types. Genes from Module 1 were highly expressed in the T cell cluster and genes from Module 2 were enriched to myeloid cells. After re-clustering of all immune cells from the two immune clusters, we identified seven immune cell clusters (**c**). Plot (**d**) showing the genes from Module 1 was enriched to T cell clusters and genes from Module 2 was enriched to myeloid cell clusters. Further re-clustering of T cell cluster (**g**) and myeloid cell cluster (**e**) respectively identified that genes from Module 1 were enriched to CD8/CTL effectors and CD4/ Treg cells (**h**), while genes from Module 2 were enriched to macrophages (**f**). CTL cytotoxic T lymphocyte, DC dendritic cell, pDCs plasmacytoid dendritic cells, NK nature killer cell, Treg regulatory T cell.

## Discussion

In this study, we integrated multi-omics data to assess the ability of a combination of genetic, epigenetic, and environmental factors to predict allergic disease. We hypothesized that multi-omics integration would provide a more accurate allergy prediction model, compared to using a single omics layer. However, our study has shown that nasal DNA methylation alone is a strong and valuable biomarker for diagnosing allergic disease. Thus, we provide a proof-of-concept that DNA methylation can be used to generate a simple diagnostic test for a complex disease such as an allergy. We extended this finding by showing that nasal DNA methylation is able to differentiate symptomatic from asymptomatic IgE sensitization, providing for the first time a set of biomarkers that can distinguish these two conditions, as well as reflecting allergic disease comorbidity. We mapped the changes in these three CpG sites to an increase in T cells and macrophages in the nasal mucosa. We propose that nasal DNA methylation accurately reflects allergic inflammation in the nasal mucosa, driven by activated T cells and macrophages.

The allergic disease often starts at an early age and may manifest in different organs, such as the respiratory system (asthma, rhino-conjunctivitis), the skin (atopic eczema), and the gastrointestinal tract (food allergy). These allergic diseases may be present at the same time in an individual (comorbidity) or may have a temporal sequence, such as displayed in the allergic march (from eczema to asthma and hay fever)[4,29]. Unsupervised statistical techniques performed in large population-based cohorts suggested that two main clusters could be observed: unaffected children and children with allergic comorbidity[30]. This pattern of extensive comorbidity of allergic diseases was further supported by the identification of genetic and epigenetic variations that were associated with all three diseases (asthma,

rhinitis, and eczema)[8,31]. Part, but not all, of this overlap is explained by IgE sensitization[4], suggesting that there are more pathogenic mechanisms shared between allergic diseases. These findings prompted us to investigate allergic disease as one unified phenotype.

One of our key findings is that DNA methylation of nasal brushings is most predictive of allergic disease. This can be explained by the fact that epigenetic signatures in the upper airways may reflect both genetic and environmental factors, in addition to cell-type composition and cell activation[32]. The three CpG sites that we use in our parsimonious model are related to genes that were enriched in immune cell pathways and mapped to T cells and macrophages. However, a limitation of our single-cell annotation is the lack of eosinophils in the nasal single-cell RNA-seq dataset, as these cells are selectively missed due to their high RNase content. Our findings indicate that the influx of immune cells into the nasal mucosa can be detected using nasal brushes and provides a strong and distinctive DNA methylation signal in allergic disease. In addition, we found that DNA methylation might mediate the effect of genetic variants on allergic disease. Taken together, these results indicate that our CpG sites may capture both genetic contribution and inflammation in the nose.

In contrast to the nasal brushings, methylation levels of CpG sites in blood did not improve the performance of our prediction model. One explanation might be that the signal in the blood is diluted due to the complex mixture of immune cells present in the sample. We previously reported a similar observation in childhood asthma, showing that the DNA methylation signal associated with asthma was stronger in purified eosinophils than in whole blood samples[12,33]. Another explanation could be that the tissue-resident T cells and macrophages may be different from their counterparts in the blood, which do not have the same methylation levels at these three CpG sites. In support of this, methylation at the three CpG sites in blood showed no obvious differences between participants with and without allergy (Supplementary Fig. 8).

Along with making an accurate prediction model, we also aimed to better understand the function of the selected CpG sites in allergic disease. In this study, allergic disease phenotype was defined by the combination of IgE sensitization and parent-reported symptoms following the definitions of the Mechanisms of the Development of ALLergy (MeDALL) consortium[4], which takes both IgE-mediated mechanisms and multimorbidity into consideration. By creating subsets of participants based on phenotype status, we could examine the function of the three CpG sites on IgE sensitization and multimorbidity. Although IgE sensitization is generally used to define the allergic status, sensitization to allergens does not necessarily mean the participants have allergy symptoms[34]. Asymptomatic allergic participants may differ from symptomatic participants in total serum IgE levels, mono- or polysensitization, presence of Treg cells, and basophil reactivity[35]. Therefore, we need effective biomarkers that can distinguish asymptomatic sensitization from allergic disease. In our study, the nasal DNA methylation levels of three CpG sites were lower in symptomatic than asymptomatic IgE-sensitized participants, showing the potential of these three CpG sites in nasal epithelial cells to be a biomarker for the presence of symptoms in IgE-sensitized individuals, or in other words, to be a more selective biomarker for allergic disease. Thus, these CpG sites are not simply a nasal reflection of the presence of specific IgE, but they also capture the presence of symptomatic disease. In addition, the association of methylation levels at these three CpG sites and the number of allergic symptoms (asthma/ rhinitis/ eczema) strongly suggest that our findings could also be used as a biomarker for allergic multimorbidity, adding more clinical relevance to our work. These findings were strongly replicated in a population of Hispanic children, showing that our prediction model is valid across at least two ethnicities.

Contrary to the predictive power of DNA methylation, genetics did not contribute to the performance of our model. At first sight,

inherent genetic variants show great potential for early diagnosis of various diseases, including obesity, coronary artery disease, and Alzheimer's disease[16,36,37]. Large genome-wide association studies (GWAS) have identified many genetic variants associated with allergic disease, which enables allergic disease to be predicted based on polygenetic risk scores. However, significant genetic variants identified by a recent GWAS only explained a small proportion of disease heritability (3.2% for asthma, 3.8% for hay fever, and 1.2% for eczema)[8], and these identified variants were mostly common variants with relatively small risk effects and ones that tend to be difficult for correctly identifying risk at the individual level, thus offering limited predictive power for allergic disease[38]. Dijk et al.[39] generated a prediction model for asthma during the first 8 years of life based on environmental factors and genetic risk scores in 1858 children of the PIAMA study. Genetic risk scores based on SNPs identified by a large asthma GWAS did not add predictive power for asthma over using just family history and environmental factors. This was also shown in an independent Swedish cohort study.

While our three-CpG model can predict the presence of an allergy, several limitations should be taken into consideration. Firstly, our model was trained in Dutch adolescents with a narrow age range. For the two cohorts of younger children (COPSAC2010 and MAKI), model extrapolation resulted in a low recall rate, in contrast to good replicability in the independent adolescent cohort. This may be explained by changes in DNA methylation with aging[40]. From this, we conclude that the current model is likely age-dependent and best applicable to adolescents. Further studies should address if these, or other CpG sites, can accurately diagnose allergies in younger children. Secondly, as our model was trained using cross-sectional data, we could not assess if our model could predict future disease status. Finally, our findings that genetic variants did not contribute to the prediction of allergic disease might be due to our relatively small dataset. However, it has been reported that also in larger datasets, genetic risk variants of allergic sensitization had limited value for clinical prediction at an individual level[41]. Although we have shown the prediction power of three nasal CpG sites, further mechanistic work is needed to disentangle the causal pathway from SNPs, CpG methylation, to disease status.

This study has three implications for biomarker discovery. Firstly, on a general level, the combination of nasal methylation sites is suggested to be a strong biomarker[42] that is replicable across different ethnicities. Secondly, there are promising implications for the field of pediatric allergy: physicians are looking for accurate non-invasive methods to diagnose diseases, especially in young children, and to reduce misdiagnoses[43]. Nasal DNA methylation biomarkers would only require a simple nasal swab rather than a blood test to assist physicians in making a diagnosis. In addition, our model's parsimonious nature– with only three CpGs– makes it more cost-effective and shows the potential to distinguish symptomatic and asymptomatic sensitization and disease comorbidity. We do acknowledge, however, that further validation in independent populations is required to assess the clinical utility of the predictive model. Thirdly, epigenetics biomarkers hold encouraging implications for the future. Besides diagnosis, these might be used to facilitate decisions on personalized medicine and treatment selection, though also for this, the model still needs to be validated in other cohorts' data. In cancer research, this has already proved to be feasible and valuable[44].

In conclusion, we have demonstrated that nasal DNA methylation has good prediction power for allergic diseases. We have presented a parsimonious model using only three nasal CpG sites that can predict allergic disease in adolescents; it was successfully replicated in an independent adolescent cohort. We show that nasal methylation bears information on the presence of allergic disease in the presence of IgE sensitization, and also reflects allergic disease multimorbidity. These CpG sites reflect the influx of T cells and macrophages that contribute to allergic inflammation.

## Methods

### Study and population description

The discovery analysis was performed in the PIAMA birth cohort. Details of the cohort have been published previously[22]. The PIAMA study started in 1996 with 3963 newborns in the Netherlands, and the current study used the data from a 16-year follow-up. We completed the medical examinations of in total 802 children at age 16 years, and 348 individuals had complete data of their clinical phenotype at 16 years old, genotype data, blood and nasal DNA methylation data, and environmental exposure data −these were all included in the discovery analysis of model generation. Characteristics of the dataset used in this analysis compared to the initial PIAMA dataset can be found in Supplementary Table 12.

In this study, an allergic disease case was defined as a child with at least one of three allergic diseases (asthma/rhinitis/eczema) in combination with the presence of IgE for at least one allergen. Details on allergic disease definitions, specific IgE measurements, and study questionnaires can be found in the online supplementary materials. Detailed numbers of participants with each disease and IgE sensitization in PIAMA can be found in Supplementary Fig. 9.

### Ethical approval

The Medical Ethical Committees of the participating institutes approved the study (Utrecht and Groningen METC (*Medisch Ethische Toetsings Commissie*) protocol number 12−019/K), and the parents and legal guardians of all participants, and later the participants themselves, gave written informed consent.

### Data measurement and quality control

**Genotype data.** DNA was extracted from whole blood samples and nasal brushing samples, which were collected from the lower inferior turbinate. Genotyping was performed in four phases using four different platforms, Illumina Human610 quad array, Illumina HumanOmniExpress array, Illumina Human Omni Express Exome Array, and Illumina Infinium Global Screening Array. Quality control (QC) of each phase was performed and then the data were merged together. SNPs were harmonized by base pair position annotated to genome build 37. In total, 2075 individuals remained after quality control, and their data from the four platforms were merged together. Then imputation was performed using the Michigan server with a reference panel of HRC.r1. SNPs of high quality (imputation quality score Rsq >0.8, MAF >0.01, and HWE $<1 \times 10^{-12}$) were used for further analysis. After stringent QC, 2075 samples and 2,893,496 SNPs remained. Quality control on the genotype sequencing data was performed using plink (1.07) and VCFtools (0.1.12b) was used to get the SNP dosages.

**DNA methylation data.** Blood and nasal DNA methylation were measured by Infinium HumanMethylation450 BeadChip array. QC steps are described in the supplementary methods and were published previously[31]. After QC, 613 blood samples and 478 nasal samples with 436,824 CpG probes remained. Methylation β-values at a given CpG were derived from the ratio of the methylated probe intensity to overall intensity (sum of methylated and unmethylated probe intensities). Then β-values were transformed to methylation values (M-values) as $\log_2(\beta/(1-\beta))$, which were used in the downstream analysis.

**RNA-seq data.** Total RNA was also extracted from nasal brushing samples and was sequenced by Illumina HiSeq2500 sequencer using default parameters for paired-end sequencing (2 × 100 bp). Details on the QC and alignment steps are disclosed in the supplement and were reported previously[31]. In total, 17,156 expressed features and 326 samples passed QC. After matching with DNA methylation data, 244 samples remained for downstream analysis. Raw count data were transformed to $\log_2$CPM using the *voom* function in the R package *limma*[45].

### Modeling approach

**Feature selection.** For feature selection, we selected an initial list of promising candidate variables based on previous research. The features included:

**Environmental factors.** Perinatal/environmental factors were collected using questionnaires during pregnancy and the first years of life (pets during pregnancy, maternal smoking, breastfeeding, older siblings at home, and low birth weight[46,47]).

**Genetic factors.** About 136 SNPs were chosen based on their strong association with the allergy phenotype in a GWAS[8], of which 101 passed our QC thresholds. Moreover, the polygenic risk score (PRS) for allergy was calculated from 4813 allergy-associated SNPs[8], as well as PRSs based on 660 asthma SNPs[48], 8 rhinitis SNPs[49], 425 eczema SNPs[50], and 221 sensitization SNPs[41].

**Nasal and blood DNA methylation.** The nasal and blood CpG sites were pre-selected based on EWAS summary statistics [FDR <0.05][12,31,33]. After excluding CpGs, which were not available in the discovery and replication cohorts, we included 134 nasal and 219 blood CpG sites in the analyses.

**Sex and age.** In total, 348 individuals had complete data on all the above features and were included in the model training.

All candidate features can be founded in Supplementary Data 2, as well as their study reference in Supplementary Table 13.

### Machine learning method selection

We examined six supervised machine learning methods, including XGBoost[51], Random Forest[52], Support Vector Machine[53], Naive Bayes[54], Neural Network[55], and Elastic Net[23]. Hyperparameter tuning was performed using Grid Search with ROC AUC as an evaluation metric, whereby each hyperparameter combination was evaluated using repeated cross-validation. Based on these results, we selected the method that provided accurate test performance, low overfitting, and good interpretability. The selected model was used for downstream analysis.

### Model training

Models were estimated within a cross-validation framework, and ten-times repeated tenfold cross-validation was adopted. This allowed for better model assessment, as it canceled out the randomness in fold splits and gave insight into performance variation over various runs[56]. To counteract the imbalance of the data (~20% cases and 80% controls), different sampling techniques were assessed, including SMOTE[57], downsampling, and upsampling. The final model included upsampling in the training data, such that both classes had the same frequency, by adding additional samples to the minority class with replacement. Test performance was defined as the average over the evaluation metric for the repeated cross-validation predictions, whereas model stability was assessed by inspecting the variation in performance of overall model runs. Besides the ROC curve, the precision-recall curve (PRC) was used as an additional means for evaluating the model, to avoid the potentially misleading interpretation of ROC when working with imbalanced data[25]. Model training and evaluation was performed using the *caret* (6.0.86) package in R (3.5.1)[58].

### Predictive contribution of each data layer

The predictive contribution of each data layer is defined as the uplift in prediction performance after adding all its respective features to the model. We adhered to an iterative procedure, where layers were added sequentially. As the order of inclusion will affect the contribution found (because information can already be embedded in other data layers), we applied a sensitivity analysis on the order of the sequential addition.

## Generation of a parsimonious model

A more parsimonious model with fewer variables can lead to less overfitting and better prediction power. Firstly, the feature importance of all repeated cross-validation (cv) model runs was retrieved and the rank product statistic was used to aggregate this into a single feature with importance ranking based on all model runs[59]. Subsequently, we created a model with only the feature with the highest rank product statistic and its test performance was evaluated using ten-times repeated tenfold cross-validation. The next most important feature was added iteratively to the model until no substantial further improvement in test performance was observed. Lastly, using the "corrected repeated k-fold cv test"[24], the performance difference between incrementally more complex models was tested for significance. The final model was selected when the addition of more variables led to no significant improvement in model performance.

The 3-CpG model was also compared to the previously published 30-CpG nasal sites model that performed well in predicting atopy in the EVA-PR cohort[14]. We employed an Elastic Net model, including their prediction panel of 30-CpG sites, and made estimates with equal model specifications in terms of tuning and validation. A comparison was performed using the "corrected repeated k-fold cv test"[60].

## External replication

After validating the reduced-feature model on the PIAMA cohort, we used the Epigenetic Variation and Childhood Asthma study in Puerto Ricans study (EVA-PR, including participants aged 9 to 20 years)[14] for replication, as well as two cohorts of younger children (mean age 6 years) COPSAC2010[61,62] and MAKI[63]. Details of these cohorts are shown in the supplementary materials.

## Sensitivity analysis

To further assess the predictive power of the genetic factors, different significance thresholds for the selection of SNPs in the PRS construction were evaluated. This analysis was performed in a larger dataset of 675 individuals from the PIAMA cohort for whom both genotype and allergy phenotype data were available (see supplementary materials). Firstly, four different PRSs were calculated using SNPs selected by different $p$ value thresholds in the allergy GWAS[8] ($P < 1 \times 10^{-5}$, $1 \times 10^{-6}$, $1 \times 10^{-7}$, and $5 \times 10^{-8}$). Next, the prediction performance of each PRS was evaluated. Model performance was also examined for different subgroups of individuals stratified by IgE sensitization, for an allergy definition that is independent of sensitization and for sensitization itself as a target variable. We stratified the samples by IgE sensitization and symptom. Significant differences between each group pair were assessed using a Student's $t$-test.

## Biological interpretation of three CpG sites

To understand the function of the three CpG sites, we first annotated the CpG sites by position and then correlated the DNA methylation level to the gene expression level by eQTM analysis. Briefly, 244 participants had matched nasal DNA methylation and nasal gene expression data that were used for the eQTM analysis. We performed linear regression analysis to assess the association of the three CpG sites with all genes available, and the model was: gene expression level ~ DNA methylation levels + age + sex + batch + study center. We controlled the FDR at 0.05 after multiple testing at each CpG level. We then performed WGCNA[27] on the genes identified by eQTM analysis. Gene modules identified from WGCNA were used for KEGG pathway enrichment analysis using the R package *topGO* (https://bioconductor.org/packages/release/bioc/html/topGO.html).

Single-cell RNA-seq data of nasal brushing samples were available from four participants with asthma and five healthy controls, and were measured by 10x genomics. The detailed sample collection, alignment, QC, clustering and annotation steps have been reported previously[31]. In brief, cells with high percent.mito (>25%) and low gene counts (<500) were removed at each sample level. Ambient RNA was corrected using FastCAR (https://github.com/LungCellAtlas/FastCAR), and Scrublet[64] was used for identifying doublets. Downstream analyses, including normalization, scaling, clustering of cells, and identifying cell marker genes were performed using the R package Seurat version 4.0[65] (https://satijalab.org/seurat/). Harmony[66] was used to integrate data from different donors.

In total, ten cell-type clusters (basal, goblet 1, goblet 2, squamous, cycling, ciliated, deuterosomal, ionocytes, myeloid, and T cells) were identified. Re-clustering of the immune cell subset (myeloid and T cells) identified the presence of two T cell clusters (CTL and T cell cluster 2), three myeloid cell clusters (myeloid 1, myeloid 2, and macrophages), B cells, plasma cells, and mast cells. To identify the sub-cluster of T cells and myeloid cells, we further re-clustered the two T cell clusters and the three myeloid cell clusters, respectively. Three T cell clusters (CD8/ CTL effectors, CD4/ Treg, and NK cells) and five myeloid cell clusters (DC-1, DC-2, macrophages, pDCs, and classical monocytes) were identified. We then checked the expression pattern of eQTM genes from Module 1 and Module 2 (identified by WGCNA) in the identified cell types and immune cell subsets, by plotting the scaled average expression of the genes in the different cell types.

Methylation quantitative trait loci (MeQTLs) analysis was performed using a linear model with R package *MatrixEQTL*[67]. In total, 433 participants had matched genotype and DNA methylation data that were included in the MeQTL analysis. Mediation analysis was performed with R package *Mediation*[68]. Details of the methods can be found in the online supplementary materials.

## Reporting summary

Further information on research design is available in the Nature Portfolio Reporting Summary linked to this article.

# Data availability

Nasal, blood DNA methylation, and bulk RNA-sequencing data from the discovery cohort (PIAMA) have been deposited in the European Genome-phenome Archive (EGA), which is hosted by the European Bioinformatics Institute (EMBL-EBI) and the Centre for Genomic Regulation (CRG), under accession number EGAS00001005189 and EGA accession number EGAS00001006240. Nasal Single-cell RNA-seq data have been deposited in EGA, under accession number EGAS00001006657. Single-cell RNA-seq data count table were downloaded from Supplementary Table 2 of ref. 28 paper. Raw data to generate figures and tables are available from the corresponding author with the appropriate permission from the PIAMA team and investigators upon reasonable request and institutional review board approval. The same applies to the figure's source data of the replication cohorts. The GWAS summary statistics used for the PRS can be found in the public GWAS catalog (https://www.ebi.ac.uk/gwas/home) under the following links: allergy (https://www.ebi.ac.uk/gwas/publications/29083406); asthma (https://www.ebi.ac.uk/gwas/publications/29273806); rhinitis (https://www.ebi.ac.uk/gwas/publications/30013184); eczema (https://www.ebi.ac.uk/gwas/publications/26482879); sensitization (https://www.ebi.ac.uk/gwas/publications/23817571). For the PRS analysis, human genome build GRCh37 (https://www.ncbi.nlm.nih.gov/assembly/GCF_000001405.13/) was used, while for the single-cell RNA analysis, GRCh38 1.2.0 (https://support.10xgenomics.com/single-cell-gene-expression/software/release-notes/build) (https://www.ncbi.nlm.nih.gov/assembly/GCF_000001405.26/) was used.

# Code availability

Our allergy prediction model and its code are freely available at https://github.com/GRIAC-Bioinformatics/Allergy_prediction. The model is documented and ready to be used for predicting allergic disease with the three CpG sites as the input.

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

## Acknowledgements

The PIAMA study was supported by The Netherlands Organization for Health Research and Development; The Netherlands Organization for Scientific Research; Lung Foundation of the Netherlands (with methylation studies supported by AF 4.1.14.001); The Netherlands Ministry of Spatial Planning, Housing, and the Environment; The Netherlands Ministry of Health, Welfare, and Sport; and the Ubbo Emmius Foundation. Yang Li was supported by an ERC Starting Grant (948207) and the Radboud University Medical Centre Hypatia Grant (2018) for Scientific Research. Cancan Qi was supported by a grant from the China Scholarship Council. Cheng-Jian Xu was supported by the Helmholtz Initiative and Networking Fund (1800167).

## Author contributions

C.-J.X. and G.H.K. conceived and designed the study, M.v.B. and C.Q. performed the analysis supervised by G.H.K., and C.-J.X., I.P., J.M.V., U.G., M.Bügel, Y.L., M.v.d.B., and M.C.N. supported in the conceptual design of the study and the critical interpretation of results. O.A.C. and M.Berg were involved in the collection and interpretation of the nasal single-cell RNA data. Z.X., C.-E.T.P., E.F., A.M., A.U.E., Y.J., W.C., L.B., K.B., and J.C.C. were involved in the external replications. C.-J.X, G.H.K., M.v.b., and C.Q. wrote the manuscript with inputs from all authors. All authors read and approved the manuscript.

## Funding

## Competing interests

The authors declare no competing interests.

## Additional information

[1]University of Groningen, University Medical Center Groningen, Beatrix Children's Hospital, Department of Pediatric Pulmonology and Pediatric Allergology, Groningen, The Netherlands. [2]University of Groningen, University Medical Center Groningen, Groningen Research Institute for Asthma and COPD (GRIAC), Groningen, The Netherlands. [3]MIcompany, Amsterdam, The Netherlands. [4]Division of Pulmonary Medicine, Department of Pediatrics, UPMC Children's

Hospital of Pittsburgh and University of Pittsburgh, Pittsburgh, PA, USA. [5]COPSAC (Copenhagen Prospective Study on Asthma in Childhood), Herlev and Gentofte Hospital, Copenhagen, Denmark. [6]University of Groningen, University Medical Center Groningen, Department of Epidemiology, Groningen, The Netherlands. [7]Institute for Risk Assessment Sciences, Utrecht University, Utrecht, The Netherlands. [8]University of Groningen, University Medical Center Groningen, Department of Pathology & Medical Biology, Groningen, The Netherlands. [9]Department of Human Genetics, University of Chicago, Chicago, IL, USA. [10]Section for Bioinformatics, Department of Health Technology, Technical University of Denmark, Lyngby, Denmark. [11]University of Groningen, University Medical Center Groningen, Department of Pulmonary Diseases, Groningen, The Netherlands. [12]Centre for Individualized Infection Medicine (CiiM), a joint venture between Hannover Medical School and Helmholtz Centre for Infection Research, Hannover, Germany. [13]TWINCORE, Centre for Experimental and Clinical Infection Research, a joint venture between Hannover Medical School and Helmholtz Centre for Infection Research, Hannover, Germany. [14]Department of Internal Medicine, Radboud University Medical Center, Nijmegen, The Netherlands. [15]Department of Pediatrics, Wilhelmina Children's Hospital, University Medical Centre Utrecht, Utrecht, The Netherlands. [16]Center for Translational Immunology, University Medical Centre Utrecht, Utrecht, The Netherlands. [17]Department of Gastroenterology, Hepatology and Endocrinology, Hannover Medical School, Hannover, Germany. [18]These authors contributed equally: Merlijn van Breugel, Cancan Qi. [19]These authors jointly supervised this work: Gerard H. Koppelman, Cheng-Jian Xu.
✉e-mail: Xu.Chengjian@mh-hannover.de

