## [Peer Review File · Nature Communications]

Nasal DNA methylation at three CpG sites predicts childhood allergic diseaseREVIEWER COMMENTS

Reviewer #1 (Remarks to the Author):

This manuscript builds on established cohorts of well-phenotyped children with and without allergic diseases. Using cross-platform approaches investigating gene expression and DNA methylation profiles, the author's identified a 3-gene DNA methylation signature that tracked with allergic disease status in two different cohorts. This manuscript has several strengths, including that it is well-written and well referenced, separately analyzed subjects based on disease symptoms with and without IgE sensitization, and replicated the key finding in an ethnically diverse cohort. There are a few weaknesses, including the cross-sectional nature of the research. Furthermore, although the findings appear robust and should spur future research into mechanisms of mucosal inflammation in allergy, the overlap in the methylation signature between groups is significant, and it is not clear that this will be a clinically useful test for individual patients. This section of the Discussion should be toned down. Please also include how allergic diseases were defined in the COPSAC and MAKI studies.

Reviewer #2 (Remarks to the Author):

By applying machine learning to 467 selected features of childhood allergic diseases including genome, blood and nasal methylome, environmental factors, Merlijn van Breugel et al. developed a 3-CpG prediction model with good performance stably replicated in an independent cohort. Replication on younger cohorts failed perhaps due to age difference. The 3 CpGs were then functionally characterized by linking their methylation with gene expression and with SNP data from the same cohort. The effort is important and results interesting. However, I have the following concerns that need to be addressed by the authors.

1. The nearly equally good replication performance of the 3 CpGs was achieved in a cohort of about the same age. When applied to younger cohorts, the performance was immediately poor. As the authors stated, the poor performance can be due to effect of aging (maybe too early to use the word aging here, but development). My concern here is that the performance of the 3 CpGs can be highly age-specific. Then which age group is applicable? If invalid for younger children, the usefulness of the model is highly limited as age 16 is beyond the usual definition of childhood (ages 6 to 12). If this is the case, the authors should state that the model is for adolescents, not childhood.
2. The authors compared their 3-CpG model with published a 30-CpG model. Can authors build a prediction model based on the 30 CpGs and try on their own cohorts? I am interested to see if the 30 CpGs predict in the younger cohorts.
3. In their effort for functional analysis, the 3 CpGs were correlated to 127 genes which were then analyzed using WGCNA. In my opinion, CpGs are deemed to regulate the surrounding genes like the promoter CpGs. Here, one CpG is associated with many genes. Simple correlation does not mean functional connection. Validity of such association is of concern. Furthermore, two modules were found from WGCNA. What about the statistical significance of the modules? There is no information that characterize the modules available.
4. In the analysis of meQTL, regression model was used with mediation analysis performed. This is fine. However, detailed information about the methods are missing although it is indicated that these are available in the online supplementary materials.

Reviewer #3 (Remarks to the Author):

This paper consists of multiple distinct analyses that collectively aim to describe the utility of nasal DNAm patterns to classify individuals with allergic diseases and interpret the biological significance and/or functionality of those DNAm patterns. The work builds on prior publications by the current authors and others describing nasal DNAm patterns in cohorts of individuals with allergic diseases. The first part of the manuscript compares the utility of genetic variables, environmental variables, blood DNAm, and nasal DNAm variables in classifying individuals with or without allergic diseases. The authors come up with an optimal classification method and determine the most accurate variables for classification and show accuracy across multiple cohorts. They show this 3 nasal DNAm classifier also associated with multimorbidity from allergic conditions. In a subset of their PIAMA cohort then they identify significant eQTMs for these 3 CpG sites and use that to understand their biological significance, in particular representing gene expression of immune cells vs epithelium. In a small scRNA-seq dataset from a separate adult population they show these genes are expressed in specific populations of leukocytes as opposed to epithelial cells. They then link the CpG sites to SNPs through MeQTL analyses.

The paper is interesting, though overall I have concerns that it is a fairly modest advancement on prior work in the field and specifically prior works by these authors with these same datasets. In my opinion, it also suffers from having multiple methods and messages without a single strong/coherent message. In my opinion it tries to address multiple questions and in doing so, addresses each in an incomplete fashion and I don't believe the messages tie together well in a single manuscript. It may be better developed into multiple independent manuscripts.

Major concerns:

The first results section fundamentally asks what is the accuracy of classification of individuals with or without allergic disease using selected variables from genetic, environmental, blood DNAm, and nasal DNAm data. They conclude nasal DNAm is superior to those other variables, which provide no additional classification accuracy. I am concerned the results could be significantly biased based on the manner in which the variables included in the models were selected. The authors should clearly list what those 467 variables were and from what cohorts they were derived. I did not comprehensively check all references that are cited for the source of these variables, but it appears that the nasal DNAm CpGs are derived from studies that employed these same cohorts – namely PIAMA, EVA-PR – including studies that were meta-analyses of these cohorts. In contrast, the blood DNA methylation CpG sites are from a distinct cohort from the Pregnancy and Childhood Epigenetics (PACE) consortium. Given that the DNAm variables were already shown to be significantly different between individual with and without allergic disease in these cohorts, it is no surprise they perform well as a classifier in the present analysis and in that sense could be considered overfit as they are not derived from independent data. This also calls into question the idea of using these cohorts for training and validation in the present manuscript since it seems the variables were selected from results from both cohorts previously. The fact that they perform less well in MAKI, may be indication of this lack of generalizability as this could be due to it being a separate cohort, vs an age difference.

The second results section tests the number of nasal CpGs sites that provides the best accuracy relative to the number of variables. It appears that any one top variable has fairly comparable accuracy to multiple. Even a single CpG site has an ROC AUC of >0.81 and adding 2 additional is only a modest increase (fig 2C). On the one hand, this raises the same question as my previous comment, to what extent is this finding biased based on picking the top CpG from prior publications of these cohorts and thus is an overfit model? On the other hand, it questions the utility of trying to build a parsimonious prediction model. Presumably such a model has no clinical utility since it is not easy to measure and simply classifies established disease that can be fairly easily clinically diagnosed. If models with 1 vs 30 CpG sites have effectively the same classification performance, there seems little rationale for optimization of this to 3 CpG sites? Moreover, if the point of this work is to describe the biology of the differentially methylated sites (which is what is done in the following sections), subsetting to 3 CpG sites seems arbitrary and the authors would do better to functionally interpret broader patterns of differentially methylated sites.

In the next results section, the authors perform eQTL analyses in a subset of the PIAMA cohort with RNA-seq data. Here the goal seems to be to interpret the biological significance of these 3 CpG sites. The authors show they link to expression of immune cell genes as opposed to epithelial cell genes, suggesting the CpGs likely represent the presence of immune cells in the epithelial samples from atopic individuals. As above, I don't see the logic of limiting this analysis to those 3 CpG sites, as a descriptive analysis of differential methylation should not be restricted to so few results.

They go on to use scRNA-seq data from nasal samples from a small adult cohort of allergic and non-allergic individuals. While this data is nicely shown and is supportive, it is not particularly novel and could be derived from similar public scRNA-seq datasets including (PMID: 30135581, PMID: 30069046, PMID: 32726565, PMID: 32327758). The authors use this simply as supportive data for their eQTL analyses rather than trying to comprehensively analyse this scRNA-seq dataset.

Finally the authors perform focused meQTL analyses of these 3 CpGs with prespecified SNPs and report nominally significant results. It is difficult to know how meaningful these findings are, in particular the reported SNP-CpG pair is on different chromosomes (6 and 12). Furthermore, I have concerns about how the mediation analysis was performed. What do the authors mean when they say "The most significant SNP-CpG pair (rs9372120- cg20372759) was also associated with allergic disease in our PIAMA dataset." Are you referring to a prior publication, or the current, and if the current analysis, please explain how this analysis was performed and the statistical outcome? Additionally, I find the mediation analysis confusing. The authors should include the model Allergic disease \sim DNAm, which is presumably much more significant than Allergic disease \sim SNP, in which case the question should be whether the SNP mediates any of the relationship of Allergic disease \sim DNAm. Furthermore if the authors do wish to test the mediation effect of DNAm in relating Allergic disease \sim SNP (which is probably inadvisable if indeed DNAm has a much stronger relationship with allergic disease than SNP does) I think the linear model performed here is backwards. To test the mediation of DNAm on SNP, I believe model 2 should be SNP \sim DNAm.

Minor comments:

In addition to these major concerns I noted a number of typos (eg order of table S5, text in S11). I also think some of the discussion is overstated, eg line 314-315: "Therefore, we urgently need effective biomarkers that can distinguish asymptomatic sensitization from full-blown allergic disease." And eg Line 353-354: The authors suggest nasal methylation as a stable biomarker, yet just above they appropriately describe the limitation of their study is the lack of longitudinal data and the uncertainty of this biomarker by age. So they need to be careful about implying their result is a stable biomarker as this is unknown.

Reviewer #4 (Remarks to the Author):

This is an interesting paper where the authors take a multi-omics and machine learning approach to identify factors that predict allergic disease in early life. They identify and replicate methylation at 3 CpG sites in nasal brushing as predictive for the development of co-morbid allergic disease. Allergic disease definition will not capture asthma that may be triggered by virus or pollution etc. The primary finding of their study is that DNA methylation of nasal brushings is most predictive for allergic disease. As they state, epigenetic signatures in the upper airways likely reflect cell composition, genetic, and environmental factors. They address each of these by looking at associations with genetic data and cell composition data (except eosinophils), but the major limitation is the lack of mechanistic work that connects the dots between these factors. It is not clear what leads to these CpG differences or what genes are impacted downstream. As such, the findings are largely correlative.

Point-by-point response to the reviewers' comments

REVIEWER COMMENTS

Reviewer #1 (Remarks to the Author):

This manuscript builds on established cohorts of well-phenotyped children with and without allergic diseases. Using cross-platform approaches investigating gene expression and DNA methylation profiles, the author's identified a 3-gene DNA methylation signature that tracked with allergic disease status in two different cohorts. This manuscript has several strengths, including that it is well-written and well referenced, separately analyzed subjects based on disease symptoms with and without IgE sensitization, and replicated the key finding in an ethnically diverse cohort. There are a few weaknesses, including the cross-sectional nature of the research [1]. Furthermore, although the findings appear robust and should spur future research into mechanisms of mucosal inflammation in allergy, the overlap in the methylation signature between groups is significant, and it is not clear that this will be a clinically useful test for individual patients. This section of the Discussion should be toned down [2]. Please also include how allergic diseases were defined in the COPSAC and MAKI studies [3].

R: We are grateful to the reviewer for appreciating our manuscript and his/her summary of the main strengths of our work.

[1] The reviewer is right to describe our research as cross-sectional, and in the discussion part we mention the following:

“Secondly, as our model was trained using cross-sectional data, we could not assess if our model could predict future disease status.”
(Page 16 Line 360-361 in marked version)

[2] We agree with the reviewer that clinical usefulness of the models still needs to be validated. We follow the reviewer's suggestion and toned down the clinical usefulness in the discussion as follows:

“Besides diagnosis, these might be used to facilitate decisions on personalized medicine and treatment selection, though for this, the model still needs to be validated in other cohorts' data.”
(Page 17 Line 378-380 in marked version)

[3] Following the reviewer's suggestion, we have added the allergic disease definition for the COPSAC and MAKI cohorts

In COPSAC:

“Asthma, Eczema and Rhinitis were diagnosed longitudinally by research physicians following the same standardized algorithm. An asthma diagnosis required a certain burden of troublesome lung symptoms, response to treatment with inhaled corticosteroids and relapse after withdrawal of treatment, which was reported previously (Bisgaard et al., New England Journal of Medicine, 2016). In this replication, current **asthma** by age 6 years was defined as still having asthma symptoms and needing regular treatment with inhaled corticosteroids in the previous year.

Rhinitis is defined as the presence of sneezing, runny, itchy or stuffed nose and the absence of infection. Specifically, besides absence of infection, two or more of the following symptoms experienced in more than one hour for more than two days: Runny nose, sneezing, itchy or blocked nose. If these criteria are not met, but the symptoms persist for more than one season or are relieved by antihistamines, we include them as cases. Rhinitis symptoms were evaluated at 6 years of age.

Eczema is defined as positive according to the definitions in Hannifin & Rajka. We required three major and three minor features to be regarded as a case (Hannifin and Rajka, *Acta Derm Venereol Suppl* (Stockh) 1980; 92: 44–47). Eczema symptoms were evaluated at 6 years of age.

Sensitization (sIgE) is defined as any specific IgE against the tested allergen (>0.35 IU/mL) at age six. We used inhalant allergens to define sensitization, and we measured house dust mite, cat epithelium and dander, dog dander, timothy grass, common silver birch, mugwort, and used a general test for molds, *Cladosporium herbarium*, *Aspergillus fumigatus*, *Alternaria tenuis* and tissue transglutaminase. A child was considered sensitized if it had an inhalant sensitization.

For the current study, we used the MeDALL allergy definition as was proposed by experts (Pinart et al, *Lancet Respir Med*. 2014; 2: 131-140). A child was considered to have allergy if he/she had at least one of three diseases: asthma/rhinitis/eczema, as well as inhalant sensitization IgE (>0.35 IU/mL).”

In MAKI:

“Information on asthma, rhinitis, eczema and IgE sensitization was obtained from clinical assessment at age 6 years. Asthma was defined as parent-reported wheeze or asthma medication usage and physician diagnosed asthma. Rhinitis was defined as the child having suffered from sneezing, a runny nose, or a blocked nose in the past 12 months while he / she did not have a cold. Eczema was defined as the child ever having suffered from itchy skin rash on the knee cavities/ the front of the ankles/ the neck/ inside of the elbows/ around the eyes or ears, in the last 12 months. IgE sensitization was defined as specific IgE to inhalant allergens at age 6 years. Allergic disease was defined as the presence of asthma or rhinitis or eczema AND IgE sensitization (>0.35 IU/ml).”

The above text has been added in the supplementary method on Page 8-9 Line 179-215 and Page 10-11 Line 234-250 in the marked version.

In the Method – External replication section we refer to this information:

“Details of these cohorts are shown in the supplementary materials.”
(Page 22 Line 502 in marked version)

Reviewer #2 (Remarks to the Author):

By applying machine learning to 467 selected features of childhood allergic diseases including genome, blood and nasal methylome, environmental factors, Merlijn van Breugel et al. developed a 3-CpG prediction model with good performance stably replicated in an independent cohort. Replication on younger cohorts failed perhaps due to age difference. The 3 CpGs were then functionally characterized by linking their methylation with gene expression and with SNP data from the same cohort. The effort is important and results interesting.

However, I have the following concerns that need to be addressed by the authors.

1. The nearly equally good replication performance of the 3 CpGs was achieved in a cohort of about the same age. When applied to younger cohorts, the performance was immediately poor. As the authors stated, the poor performance can be due to effect of aging (maybe too early to use the word aging here, but development). My concern here is that the performance of the 3 CpGs can be highly age-specific. Then which age group is applicable? If invalid for younger children, the usefulness of the model is highly limited as age 16 is beyond the usual definition of childhood (ages 6 to 12). If this is the case, the authors should state that the model is for adolescents, not childhood.

R: We thank the reviewer for appreciating our manuscript and providing us with extensive suggestions for further improvement. We agree with the reviewer that our model may be age specific and one important reason is that DNA methylation is age-specific. We have checked the 3 CpG sites at EWAS catalog (<http://www.ewascatalog.org>), and found that cg24224501 and cg01870976 in blood are associated with age (development) from birth to late adolescence (Mulder et al, Hum Mol Genet, 2021, PMID 33450751). This is in line with the observation that mean nasal DNA-methylation of the top CpG sites differs in the control group between the adolescent and the younger populations. As the subjects in EVA-PR are within the age range of 9-20 years, we regard our model to be effective at the age range of adolescents.

To further investigate this, we have performed an age-stratified replication in the EVA-PR cohort. (Note that this was not possible in PIAMA due to very low age variance.) The results hereof are shown below. This shows that model performance is indeed highest for the age group that is comparable to PIAMA (15-17 year old), but also performs very good at higher ages (age 18-20 years). Model performance remains satisfactory for the younger stratum of 9-14 years, which strengthens our confidence in the model's applicable age range.

We have included the above figure in the supplementary material and reference it in the manuscript:

- “We also performed the replication on stratified age groups in EVA-PR (Figure S10), which showed that model performance is very good at similar (15-17 year) and higher ages (18-20 year); and remains satisfactory for the age group 9-14 year.”
(Page 9 Line 179-181 in marked version)

However, as the MAKI and COPSAC cohorts show, the model is less effective for younger children aged 6 years. So far, we do not have data to test the usefulness of the model between age 6 to age 9, nor the adult population.

One promising and interesting approach would be to re-calibrate or re-build the model for younger populations. For this, the same modeling, optimization and replication approach could be adopted. Most likely, this would result in a different model specification with other features. While we would be highly interested in such an extension, unfortunately we don't have sufficient well-phenotyped data from this population to reliably build such a model. Looking forward, it could well be the case that different age groups could have different predictions models.

To address the reviewer's suggestions, we have explicitly mentioned the age range of the cohorts in the abstract. Moreover, we have extended our discussion on the age-dependent limitation. Specifically, we have changed the following in the discussion:

- Previous text: Therefore, extra caution is needed when extending our model to younger children and further studies should address if these, or other CpG sites, can contribute to diagnosing allergy in younger children.

- New text: “From this, we conclude that the current model is likely age-dependent and best applicable to adolescents. Further studies should address if these, or other CpG sites, can accurately diagnose allergy in younger children.”
(Page 16 Line 358-360 in marked version)

2. The authors compared their 3-CpG model with published a 30-CpG model. Can authors build a prediction model based on the 30 CpGs and try on their own cohorts? I am interested to see if the 30 CpGs predict in the younger cohorts.

R: We share that interest and thank the reviewer for suggesting this. For this, we have performed another replication to address this fruitful recommendation. We have used the original 30 CpG prediction model from Forno et al. (2018, PMID: 30584054) and applied these to the MAKI birth cohort. Below we added the ROC for their GBM and GLMNET model, together with the confusion matrix of both these models.

GLMNET model		
Reference		
Prediction	Allergy no	Allergy yes
No	0	0
Yes	230	29

GBM model		
Reference		
Prediction	Allergy no	Allergy yes
No	192	26
Yes	38	3

These results clearly show that this model does not replicate well in the younger MAKI cohort either. While consistent with our findings, performance of the more complex 30 CpG model is notably lower. It is important to state, however, that this model was trained to predict atopy which is defined as the presence of IgE sensitization, thus not allergic disease which also includes symptoms. We also replicated their model on MAKI’s atopy phenotype but obtained equally poor results (see figure below).

We propose not to include the above replication result in our manuscript, as they concern a different predictive model and hence are not part of our model's replication plan.

3. In their effort for functional analysis, the 3 CpGs were correlated to 127 genes which were then analyzed using WGCNA. In my opinion, CpGs are deemed to regulate the surrounding genes like the promoter CpGs. Here, one CpG is associated with many genes. Simple correlation does not mean functional connection. Validity of such association is of concern.

R: The concern raised by the reviewer is valid. In the functional interpretation of the 3 CpGs we have used *trans*, instead of *cis*.

The *trans* genetic regulation of methylation has been shown to overlap with interchromosomal contacts (Bonder et al, 2016, PMID: 27918535). In our opinion, *trans*-eQTM may reflect cell type presence and/or activation (Xu et al, 2018 PMID: 29496485), and may also possibly be due to the interchromosomal contacts. Besides, the *trans*-CpG sites may also play a role in regulation of gene expression according to several previous publications (Maas et al, 2020, PMID: 33092652; Yoo et al, 2015, PMID: 25569234). These papers indicated a possible regulatory effect of DNA methylation on the expression levels of genes far from the neighboring methylation site. From our previous study, we identified that *trans*-eQTM genes of asthma-associated blood CpG sites were enriched in Eosinophils (Xu et al, 2018, PMID: 29496485). Considering the above mentioned reasoning, we did not restrict our analysis to *cis*-eQTM only. However, we understand the concern of the reviewer, and based on this concern, we have performed additional analyses where we applied a *cis*-approach. We have added these results (see table below) to the supplements Table S7 eQTM.

cis-eQTM						
CpG	gene	hgnc_symbol	coef	t	P.Value	fdr
cg20372759	ENSG00000123329	ARHGAP9	-0,471	-3,281	9,87E-04	0,02
cg01870976	ENSG00000140479	PCSK6	0,396	3,335	1,19E-03	0,01

Furthermore, two modules were found from WGCNA. What about the statistical significance of the modules? There is no information that characterize the modules available.

R: For the validity of these modules and downstream analyses, we apologize for the unclear description of the method. In brief, WGCNA determines the modules according to weighted co-expression of the genes. The first step is the construction of a co-expression network. By raising the absolute value of the correlation to a power $\beta \geq 1$ (soft thresholding), the weighted gene co-expression network construction emphasizes high correlations at the expense of low correlations. Here, modules are clusters of genes with high absolute correlations. The detection of the module is the step after the construction of co-expression network, WGCNA identified modules using unsupervised hierarchical

clustering. To make it clearer to the reader, we added this text to the supplemental method in Page 13-14, Line 313-323 in marked version.

Besides, we also added the following details about the settings used in this analysis and the identified modules in the supplemental figures (Figure S8, referenced in Page 14, Line 322-323 in marked version of supplementary methods, as shown below).

- Figure S8. WGCNA analysis of eQTM genes. (a-b) Determination of soft-thresholding power in the WGCNA. (a) Analysis of the scale-free fit index for various soft-thresholding powers (β). (b) Analysis of the mean connectivity for various soft-thresholding powers. Soft threshold was selected as 9 and used in the following analysis. (c) Dendrogram of all eQTM genes clustered based on a dissimilarity measure (1-TOM), colors represent identified modules.

4. In the analysis of meQTL, regression model was used with mediation analysis performed. This is fine. However, detailed information about the methods are missing although the it is indicated that these are available in the online supplementary materials.

R: We thank the reviewer for raising this issue of missing information on this method. In the online supplementary methods, a short paragraph with explanation of this method is added:

“To evaluate what proportion, if any, of an association between SNP and asthma was mediated through changes in the DNA methylation level of the three CpG sites, we conducted mediation testing using R package mediation. The SNP list (N=75) included in this analysis was obtained from previous GWAS of allergic disease. We performed an MeQTL analysis that associated the three CpG sites with SNPs previously associated with allergic disease. For the most significant SNP- CpG pair, we performed the mediation analysis. First, we tested the total effect of SNP on allergic disease (model: Allergic disease ~ SNP). Second, we tested the effect of the independent variable (SNP) on the mediator (DNA methylation; model: DNAm ~ SNP). Third, we tested the mediator’s (DNA methylation) and independent variable’s (SNP) effect on the dependent variable (allergic disease; model: Allergic disease ~ SNP +DNAm). Lastly, we compared the direct and the indirect effect and estimated the mediation effect.”

(Page 13, Line 299-311 in marked version)

Reviewer #3 (Remarks to the Author):

This paper consists of multiple distinct analyses that collectively aim to describe the utility of nasal DNAm patterns to classify individuals with allergic diseases and interpret the biological significance and/or functionality of those DNAm patterns. The work builds on prior publications by the current authors and others describing nasal DNAm patterns in cohorts of individuals with allergic diseases. The first part of the manuscript compares the utility of genetic variables, environmental variables, blood DNAm, and nasal DNAm variables in classifying individuals with or without allergic diseases. The authors come up with an optimal classification method and determine the most accurate variables for classification and show accuracy across multiple cohorts. They show this 3 nasal DNAm classifier also associated with multimorbidity from allergic conditions. In a subset of their PIAMA cohort then they identify significant eQTMs for these 3 CpG sites and use that to understand their biological significance, in particular representing gene expression of immune cells vs epithelium. In a small scRNA-seq dataset from a separate adult population they show these genes are expressed in specific populations of leukocytes as opposed to epithelial cells. They then link the CpG sites to SNPs through MeQTL analyses.

The paper is interesting, though overall I have concerns that it is a fairly modest advancement on prior work in the field and specifically prior works by these authors with these same datasets. In my opinion, it also suffers from having multiple methods and messages without a single strong/coherent message. In my opinion it tries to address multiple questions and in doing so, addresses each in an incomplete fashion and I don’t believe the messages tie together well in a single manuscript. It may be better developed into multiple independent manuscripts.

R: We would like to thank the reviewer for assessing our manuscript and sharing his/her elaborate summary of the work. We particularly appreciate the critical depth of the raised concerns.

Major concerns:

Major 1: The first results section fundamentally asks what is the accuracy of classification of individuals with or without allergic disease using selected variables from genetic, environmental, blood DNAm, and nasal DNAm data. They conclude nasal DNAm is superior to those other variables, which provide no additional classification accuracy. I am concerned the results could be significantly biased based on the manner in which the variables included in the models were selected. The authors should clearly list what those 467 variables were and from what cohorts they were derived. I did not comprehensively check all references that are cited for the source of these variables, but it appears that the nasal DNAm CpGs are derived from studies that employed these same cohorts – namely PIAMA, EVA-PR – including studies that were meta-analyses of these cohorts. In contrast, the blood

DNA methylation CpG sites are from a distinct cohort from the Pregnancy and Childhood Epigenetics (PACE) consortium. Given that the DNAm variables were already shown to be significantly different between individual with and without allergic disease in these cohorts, it is no surprise they perform well as a classifier in the present analysis and in that sense could be considered overfit as they are not derived from independent data. This also calls into question the idea of using these cohorts for training and validation in the present manuscript since it seems the variables were selected from results from both cohorts previously. The fact that they perform less well in MAKI, may be indication of this lack of generalizability as this could be due to it being a separate cohort, vs an age difference.

R: We thank the reviewer for raising this important question. In the following table, we list the resources of the 467 candidate variables. This table is also added as Table S15 in our manuscript. The blood DNA methylation CpG sites are from the IoW cohort, MeDALL and PACE consortium study, where PIAMA is also part of the MeDALL and PACE consortium. All replication cohorts are independently treated as “black box” and we did not include any information from these cohorts in the feature selection or model training. The reason that nasal methylation can provide better prediction power is due to the larger effect size than blood methylation and genetics.

As DNA methylation is age-specific, and our model is trained on PIAMA data collected at age 16, it can be generalized to the EVA-PA cohort with children aged 9-20 years, but not to the 6-year old children in MAKI and COPSAC.

Table S15. Literature source of initial feature selection		
Daya layer	Study reference	Number of features selected (post QC)
Environment	Miliku and Azad (2018)	-
Perinatal	Murrison et al. (2019)	-
Genetics SNPs	Ferreira et al. (2017)	75
Genetics PRS - Asthma	Demenaïs et al. (2018)	660
Genetics PRS - Rhinitis	Waage et al. (2018)	8
Genetics PRS - Eczema	Paternoster et al. (2015)	425
Genetics PRS - Allergy	Ferreira et al. (2017)	4813
Genetics PRS - IgE sensitization	Bønnelykke et al. (2013)	221
Blood DNA methylation	Xu et al. (2018); Sarah et al. (2019); Zhang et al. (2019); Xu et al. (2021)	219
Nasal DNA methylation	Qi et al. (2020)	134
Caption		
We used features that were found to be associated with allergic diseases in earlier studies. Especially for the multi-omics layers, a substantial filtering of features (SNPs and CpG sites) was needed to use these sources in ML models.		
For the PRS, selected number of SNPs is based on GWAS significance threshold of 1e-07.		

Major 2: The second results section tests the number of nasal CpGs sites that provides the best accuracy relative to the number of variables. It appears that any one top variable has fairly comparable accuracy to multiple. Even a single CpG site has an ROC AUC of >0.81 and adding 2 additional is only a modest increase (fig 2C). One the one hand, this raises the same question as my previous comment, to what extent is this finding biased based on picking the top CpG from prior publications of these cohorts and thus is an overfit model? On the other hand, it questions the utility of trying to build a parsimonious prediction model. Presumably such a model has no clinical utility since it is not easy to measure and simply classifies established disease that can be fairly easily clinically diagnosed [1]. If models with 1 vs 30 CpG sites have effectively the same classification performance, there seems little rationale for optimization of this to 3 CpG sites? Moreover, if the point of this work is to describe the biology of the differentially methylated sites (which is what is done in the following sections), subsetting to 3 CpG sites seems arbitrary and the authors would do better to functionally interpret broader patterns of differentially methylated sites.

R: We thank the reviewer for his/her critical assessment of the modeling technicalities of this work.

The number of CpG sites included in the model is based on statistical testing and cross validation (corrected repeated k-fold cross-validation (cv) test as described in Bouckaert & Frank, 2004*). Based on this test, we could show that the increase from 1 to 2, and from 2 to 3 CpGs, resulted in a significant increase in ROC AUC (p<0.05), while the step from 3 to 4 was insignificant (p-value = 0.76). Therefore, the choice for 3 CpGs is optimal. As we want to functionally understand the predictors in this model, we have consciously limited the functional interpretation to these 3 CpGs.

To enhance transparency of these test results, we have added the complete table with test statistics to as Table S14. This table is also referenced in the Result section:

“... did not further improve model performance (p-value = 0.76, corrected repeated k-fold cross-validation (cv) test²⁴, Figure 2-c). The other test statistics are displayed in Table S14.”

[1] The remark about the clinical usefulness of this model as diagnostic tool was also raised by another reviewer (#1). Please find below our response to that remark:

We agree with the reviewer [#1] that clinical usefulness of the models still needs to be validated. We follow the reviewer’s suggestion and toned down the clinical usefulness in the discussion as follows:

“Besides diagnosis, they might be used to facilitate decisions on personalized medicine and treatment selection, though for this the model still needs to be validated in other cohorts’ data.”

(Page 17 Line 378-380 in marked version)

*Bouckaert, R. R., & Frank, E. (2004, May). Evaluating the replicability of significance tests for comparing learning algorithms. In *Pacific-Asia Conference on Knowledge Discovery and Data Mining* (pp. 3-12). Springer, Berlin, Heidelberg.

Major 3: In the next results section, the authors perform eQTL analyses in a subset of the PIAMA cohort with RNA-seq data. Here the goal seems to be to interpret the biological significance of these 3 CpG sites. The authors show they link to expression of immune cell genes as opposed to epithelial cell genes, suggesting the CpGs likely represent the presence of immune cells in the epithelial samples from atopic individuals. As above, I don’t see the logic of limiting this analysis to those 3 CpG sites, as a descriptive analysis of differential methylation should not be restricted to so few results [1].

They go on to use scRNA-seq data from nasal samples from a small adult cohort of allergic and non-allergic individuals. While this data is nicely shown and is supportive, it is not particularly novel and could be derived from similar public scRNA-seq datasets including (PMID: 30135581, PMID: 30069046, PMID: 32726565, PMID: 32327758). The authors use this simply as supportive data for their eQTL analyses rather than trying to comprehensively analyse this scRNA-seq dataset [2].

R: We thank the reviewer for the suggestion.

[1] For the first question about the eQTM analysis, as we mentioned in the previous comment [Major comment 2], we chose the three CpG sites based on the model with best performance. Therefore we would like to investigate why these three CpG sites included in the model can distinguish allergy/non-allergy subjects. That’s why we only focus on the three CpG sites. Following the reviewer’s suggestion, we performed additional analyses using the top 5 CpG sites to do the eQTM analysis and plotted the expression level of identified eQTM genes in each cell type. Below, we can still see the signal from T cells and Myeloid cells, and at the same time we also see the signal from epithelial cells such as ciliated cells (see Figure below). This was consistent with our previous finding (Qi et al, 2020, PMID: 31953105), where we found that eQTM genes related to nasal CpG sites, which were associated with asthma and/or rhinitis, are also marker genes of ciliated and immune cells. Considering the flow of our paper, we propose to present our results restricted to the three CpG sites.

[2] Following the reviewer’s suggestion, we used new single cell data from the suggested paper (Ordovas-Montanes et al, 2018, PMID: 30135581) to test whether we can get consistent results using a different dataset. We made the following plot, where we plotted the genes from the two modules that we identified, in the cell types that were identified by Ordovas-Montanes *et al*, and the results are consistent with what we found in our own dataset: we saw the enrichment of genes from module 1 in T cells and genes from module 2 in Myeloid cells.

We have added this figure to the supplements (Figure S9) and added one sentence in our paper:

“The findings were consistent when using another nasal scRNAseq dataset²⁸ (FigureS9) which used the nasal polyp tissues that were slightly different from nasal brushing cells used in this study.”

(Page 11 Line 240-234 in marked version)

Major 4: Finally the authors perform focused meQTL analyses of these 3 CpGs with prespecified SNPs and report nominally significant results. It is difficult to know how meaningful these findings are, in particular the reported SNP-CpG pair is on different chromosomes (6 and 12). Furthermore, I have concerns about how the mediation analysis was performed. What do the authors mean when they say “The most significant SNP-CpG pair (rs9372120- cg20372759) was also associated with allergic disease in our PIAMA dataset.” Are you referring to a prior publication, or the current, and if the current analysis, please explain how this analysis was performed and the statistical outcome?

Additionally, I find the mediation analysis confusing. The authors should include the model Allergic disease ~ DNAm, which is presumably much more significant than Allergic disease ~ SNP, in which case the question should be whether the SNP mediates any of the relationship of Allergic disease ~ DNAm. Furthermore if the authors do wish to test the mediation effect of DNAm in relating Allergic disease ~ SNP (which is probably inadvisable if indeed DNAm has a much stronger relationship with allergic disease than SNP does) I think the linear model performed here is backwards. To test the mediation of DNAm on SNP, I believe model 2 should be SNP ~ DNAm.

R: We thank the reviewer for this comment and apologize that we did not express ourselves clearly in the method of the mediation analysis. Firstly, about the SNP-CpG pairs on different chromosome, here we selected the allergy-associated CpG sites from published paper (Ferreira, M. A. et al. Nat. Genet. (2017)), and then correlated these SNPs with the three CpG sites in our PIAMA dataset. And yes, we actually did the trans-MeQTL analysis. From previous publications (Bonder et.al. Nat. Genet. 2016), we learned that there are a lot of *trans*-MeQTLs identified, and showed that “disease-associated variants have widespread effects on DNA methylation in trans that likely reflect differential occupancy of trans binding sites by *cis*-regulated transcription factors.” Taking this into consideration, we did not restrict the analysis to *cis*-MeQTL.

Secondly, as mentioned previously, the SNPs were selected based on a previous GWAS, then we also would like to test if the association exists in our PIAMA dataset, which is also the first step of our mediation analysis. As for the mediation analysis, we used the R package mediation, according to the paper Tingley et al “Mediation: R package for causal mediation analysis”, we would like to check if the effect of SNP on allergy is mediated by DNA methylation or not. Here the outcome is allergy and the mediator is a CpG site. To make this part clearer, we added more details in the method and supplements part:

“To evaluate what proportion, if any, of an association between SNP and asthma was mediated through changes in the DNA methylation level of the three CpG sites, we conducted mediation testing using R package mediation. SNP list included in this analysis was obtained from previous GWAS of allergic disease. We performed a MeQTL analysis that associated the three CpG sites with SNPs previously associated with allergic disease. For the most significant SNP- CpG pair, we performed the mediation analysis. First, we tested the total effect of SNP on allergic disease (model: Allergic disease ~ SNP). Second, we tested the effect of the independent variable (SNP) on the mediator (DNA methylation; model: DNAm ~ SNP). Third, we tested the mediator’s (DNA methylation) and independent variable’s (SNP) effect on the dependent variable (allergic disease; model: Allergic disease ~ SNP +DNAm). Lastly, we compared the direct and the indirect effect and estimated the mediation effect.”

(Page 13, Line 299-311 in marked version)

Minor comments:

In addition to these major concerns I noted a number of typos (eg order of table S5, text in S11). I also think some of the discussion is overstated, eg line 314-315: “Therefore, we urgently need effective biomarkers that can distinguish asymptomatic sensitization from full-blown allergic disease.” And eg Line 353-354: The authors suggest nasal methylation as a stable biomarker, yet just above they appropriately describe the limitation of their study is the lack of longitudinal data and the uncertainty of this biomarker by age. So they need to be careful about implying their result is a stable biomarker as this is unknown.

R: The textual suggestions from the reviewer are appreciated and we have revised the manuscript’s text accordingly. Also, the referenced parts from the discussion have been toned down. In regard to the last remark, we have added a part to the discussion where we state that our model is indeed age-specific. (This was also raised by reviewer #1).

“From this, we conclude that the current model is likely age-dependent and best applicable to adolescents. Further studies should address if these, or other CpG sites, can accurately diagnose allergy in younger children.”

(Page 16 Line 358-360 in marked version)

Reviewer #4 (Remarks to the Author):

This is an interesting paper where the authors take a multi-omics and machine learning approach to identify factors that predict allergic disease in early life. They identify and replicate methylation at 3 CpG sites in nasal brushing as predictive for the development of co-morbid allergic disease. Allergic disease definition will not capture asthma that may be triggered by virus or pollution etc. The primary finding of their study is that DNA methylation of nasal brushings is most predictive for allergic disease. As they state, epigenetic signatures in the upper airways likely reflect cell composition, genetic, and environmental factors. They address each of these by looking at associations with genetic data and cell composition data (except eosinophils), but the major limitation is the lack of mechanistic work that connects the dots between these factors. It is not clear what leads to these CpG differences or what genes are impacted downstream. As such, the findings are largely correlative.

R: We are grateful for the reviewer’s appreciation of our manuscript and the synopsis of most relevant findings. We agree with the reviewer that it’s important to connect the links between different molecular layers, which will be very meaningful for biological interpretation. In this paper, we are aware of that and we did invest effort to find some connections, by linking CpG to gene expression (eQTM) and genetic risk variants (MeQTL), and we did find that cg20372759 is a mediator of the SNP rs9372120 effect from allergy-associated SNP on allergic disease. At the same time, we have to admit that we do not have enough power to identify the full pathway from SNP, CpG methylation, gene expression, cell counts to disease. Thus further efforts are still needed.

Following the reviewer’s suggestion, we have added the following part as one limitation in our discussion.

“Although we have shown the prediction power of three nasal CpG sites, further mechanistic work is needed to disentangle the causal pathway from SNPs, CpG methylation to disease status.”
(Page 17 Line 365-367 in marked version)

REVIEWERS' COMMENTS

Reviewer #1 (Remarks to the Author):

The authors have adequately addressed my criticisms and concerns.

Reviewer #2 (Remarks to the Author):

I am satisfied by the responses to my comments and with the revision.

Reviewer #3 (Remarks to the Author):

This manuscript explores the utility of nasal DNAm patterns (vs other factors) to classify individuals with allergic diseases and interpret the biological significance and/or functionality of those DNAm patterns. Since the original version, the authors have made a number of changes that do bring important clarity to parts of the manuscript. Fundamentally, my original main concern largely remain the same as I will discuss below. That said, the work is otherwise sound and while I view it as a rather incremental step from prior publications involving these data, it seems the other reviewers view it as a more significant advancement, and as such, I would not want to hold it back from publication.

Major concerns:

My original major concern was that: the first results section fundamentally asks what is the accuracy of classification of individuals with or without allergic disease using selected variables from genetic, environmental, blood DNAm, and nasal DNAm data. The variables were pre-selected, and, notably, used nasal DNAm sites that have already been shown to be very significantly different in both the PIAMA and EVA-PR cohorts (Forno et al, The Lancet Respiratory Medicine 2019) – or at least that is my understanding. Therefore, by definition picking a classifier from variables already proven to be markedly different will lead to a highly accurate model that is “overfit” to these cohorts. True validation would have to come from a cohort that has not already been used to identify the differentially methylated regions used in the first place.

I accept the authors’ response, which effectively states why nasal DNAm were selected by the model and not other variables, and in a sense that is meaningful in showing that nasal DNAm is better at distinguishing disease than the other variables. However, the work does not really prove the accuracy of the nasal DNAm sites in a general sense. As such, I still take issue with the manuscript’s strong focus on the prediction accuracy and clinical utility, which I don’t think have been adequately demonstrated in this work. My general recommendation would be to tone down language around the generalizability and clinical utility of the results and focus more on their mechanistic insights and relative importance vs other variables – eg SNPs, blood DNAm, which the authors have shown more convincingly.

The authors have adequately addressed my other prior concerns and I appreciate all their details responses.

Reviewer #4 (Remarks to the Author):

My comments remain but they have now clearly acknowledged the limitation.

Point-by-point response to the reviewers' comments

REVIEWER COMMENTS

Reviewer #1 (Remarks to the Author):

The authors have adequately addressed my criticisms and concerns.

Reviewer #2 (Remarks to the Author):

I am satisfied by the responses to my comments and with the revision.

Reviewer #3 (Remarks to the Author):

This manuscript explores the utility of nasal DNAm patterns (vs other factors) to classify individuals with allergic diseases and interpret the biological significance and/or functionality of those DNAm patterns. Since the original version, the authors have made a number of changes that do bring important clarity to parts of the manuscript. Fundamentally, my original main concern largely remain the same as I will discuss below. That said, the work is otherwise sound and while I view it as a rather incremental step from prior publications involving these data, it seems the other reviewers view it as a more significant advancement, and as such, I would not want to hold it back from publication.

Major concerns:

My original major concern was that: the first results section fundamentally asks what is the accuracy of classification of individuals with or without allergic disease using selected variables from genetic, environmental, blood DNAm, and nasal DNAm data. The variables were pre-selected, and, notably, used nasal DNAm sites that have already been shown to be very significantly different in both the PIAMA and EVA-PR cohorts (Forno et al, The Lancet Respiratory Medicine 2019) – or at least that is my understanding. Therefore, by definition picking a classifier from variables already proven to be markedly different will lead to a highly accurate model that is “overfit” to these cohorts. True validation would have to come from a cohort that has not already been used to identify the differentially methylated regions used in the first place.

I accept the authors' response, which effectively states why nasal DNAm were selected by the model and not other variables, and in a sense that is meaningful in showing that nasal DNAm is better at distinguishing disease than the other variables. However, the work does not really prove the accuracy of the nasal DNAm sites in a general sense. As such, I still take issue with the manuscript's strong focus on the prediction accuracy and clinical utility, which I don't think have been adequately demonstrated in this work. My general recommendation would be to tone down language around the generalizability and clinical utility of the results and focus more on their mechanistic insights and relative importance vs other variables – eg SNPs, blood DNAm, which the authors have shown more convincingly.

The authors have adequately addressed my other prior concerns and I appreciate all their details responses.

R: We thank the reviewer for appreciating our revised manuscript and elucidating his/her concerns regarding the prediction accuracy in further detail. We acknowledge that the pre-selection of nasal methylation sites was based on earlier work which was partly performed in the PIAMA cohort (Qi *et al.* (2020)). However, this selection was based purely on analysis on the PIAMA cohort and did not involve any data from the EVA-PR cohort. In that sense, we did not use information from the replication cohorts in our primary selection.

Following the reviewer's recommendations, we have toned down the language in our manuscript in various places:

- Abstract (line 73):
Previous text: Our study offers a simple, non-invasive diagnostic test for childhood allergy.
New text: Our study offers a proof of concept for methylation-based allergy diagnosis.
- Discussion (line 379):
New text: We do acknowledge, however, that further validation in independent populations is required to assess the clinical utility of the predictive model.

Reviewer #4 (Remarks to the Author):

My comments remain but they have now clearly acknowledged the limitation.